# A population-based temporal logic gate for timing and recording chemical events

Victoria Hsiao[1,*], Yutaka Hori[2], Paul WK Rothemund[3] & Richard M Murray[1]

## Abstract

Engineered bacterial sensors have potential applications in human health monitoring, environmental chemical detection, and materials biosynthesis. While such bacterial devices have long been engineered to differentiate between combinations of inputs, their potential to process signal timing and duration has been overlooked. In this work, we present a two-input temporal logic gate that can sense and record the order of the inputs, the timing between inputs, and the duration of input pulses. Our temporal logic gate design relies on unidirectional DNA recombination mediated by bacteriophage integrases to detect and encode sequences of input events. For an *E. coli* strain engineered to contain our temporal logic gate, we compare predictions of Markov model simulations with laboratory measurements of final population distributions for both step and pulse inputs. Although single cells were engineered to have digital outputs, stochastic noise created heterogeneous single-cell responses that translated into analog population responses. Furthermore, when single-cell genetic states were aggregated into population-level distributions, these distributions contained unique information not encoded in individual cells. Thus, final differentiated sub-populations could be used to deduce order, timing, and duration of transient chemical events.

**Keywords**  DNA memory; event detectors; integrases; population analysis; stochastic biomolecular models

**Subject Categories**  Quantitative Biology & Dynamical Systems; Synthetic Biology & Biotechnology

**Mol Syst Biol. (2016) 12: 869**

## Introduction

Engineered bacteria could one day be powerful self-replicating biosensors with environmental, health, and industrial applications. Synthetic biology has made important strides in identifying and optimizing genetic components for building such devices. In particular, much work has focused on Boolean logic gates that detect the presence or absence of static chemical signals (Gardner *et al*, 2000; Anderson *et al*, 2007; Wang *et al*, 2011; Moon *et al*, 2013; Shis *et al*, 2014) and compute a digital response.

Temporal logic gates, which process time-varying chemical signals, have been much less explored. Pioneering work by Friedland *et al* (2009) used serine integrase-based recombination for the counting and detection of sequential pulses of inducers. But thus far, no work has studied the potential for temporal logic gates to provide information about the duration of a signal, or the time between two chemical events. Here, we present a temporal logic gate that allows us to infer analog signal timing and duration information about the sequential application of two inducer molecules to a population of bacterial cells.

Similar to previous temporal logic gates, our design takes advantage of the irreversibility of serine integrase recombination. While bistable switches have been successfully deployed as memory modules in genetic circuits (Kotula *et al*, 2014), such switches require constant protein production to maintain state and are sensitive to cell division rates and growth phase. The large serine integrases, on the other hand, reliably and irreversibly flip or excise unique fragments of DNA (Yuan *et al*, 2008). Thus, logic circuits built from integrases intrinsically include DNA-level memory that requires virtually no cellular resources to maintain state, thus enabling permanent and low-cost genetic differentiation of individual bacterial cells based on transient integrase induction. Further advantages of the serine integrases include the short length (40–50 bp) and directionality of their attachment sites. Serine integrases recognize flanking DNA binding domains (attB, attP) and subsequently digest, flip or excise, and re-ligate the DNA between the attachment sites. Flipping or excision activity is determined by the relative orientation of the sites, which allows complex orientation-dependent behavior to be programmed into integrase circuits. Well-known serine integrases include Bxb1, TP901-1, and ΦC31, all of which have been used to demonstrate static-input logic gates (Bonnet *et al*, 2013; Siuti *et al*, 2013), and some have cofactors that can reverse directionality (Khaleel *et al*, 2011; Bonnet *et al*, 2012). Recently, an entirely new set of 11 orthogonal integrases was characterized, greatly expanding the set of circuits that can be built (Yang *et al*, 2014).

In contrast to previous studies of temporal logic gates, our work leverages the stochastic nature of single-cell switching to create a

1  Biology and Biological Engineering, California Institute of Technology, Pasadena, CA, USA
2  Applied Physics and Physico-Informatics, Keio University, Yokohama, Kanagawa, Japan
3  Computation & Neural Systems, California Institute of Technology, Pasadena, CA, USA
   *Corresponding author. Tel: +1 626 395 4140; E-mail: vhsiao@caltech.edu

robust population-level response to a time-varying chemical signal. The fundamental nature of living cells that makes them so attractive for engineering—their extremely low energy operation in the limit of using small numbers of molecules to represent information—is also inextricably linked to stochasticity and noise. By traditional engineering standards, synthetic circuits would ideally perform identically within every cell of a population. When this ideal is applied to biology, the stochastic nature of molecular processes, particularly at low-copy numbers, presents a significant barrier to reliable outputs from engineered cells. Thus, while natural cellular dynamics and differentiation take advantage of noisy gene expression (Elowitz *et al*, 2002; Süel *et al*, 2007), synthetic circuits often require noise reduction for proper function (Dunlop *et al*, 2008). Recent work has taken a different direction, toward understanding of population-level dynamics. This includes analysis of both stochastic cellular responses to inputs (Uhlendorf *et al*, 2012; Ruess *et al*, 2015) and changes in collective population-level memory in response to stress (Mathis & Ackermann, 2016). Such efforts suggest that a deeper understanding of the inherent heterogeneity in biological systems might eventually lead to circuit designs that operate on distributions of cellular responses, rather than depending on homogeneous responses from all cells.

It is with this vision in mind that we designed a two-input temporal logic gate using strategically interleaved and oriented integrase (Bxb1, TP901-1) DNA recombination sites and used this gate to engineer an *E. coli* strain with four possible genetically differentiated end states. This strain contains single genomic copies of the temporal logic gate, ensuring digital-yet-stochastic responses from individual cells. We then utilized the heterogeneity of individual cellular responses to encode sequences of chemical inputs into the overall population response and used a stochastic model of single-cell trajectories to predict the population response. By analyzing the distributions of final cell states, we can deduce the timing and pulse duration of transient chemical pulses and show that cumulative population-level distributions contain additional event information not encoded in any single cell. Furthermore, because the states are genetically encoded, we can recover details of a chemical event long after its occurrence.

## Results

### Design of a two-integrase temporal logic gate

We have designed a two-input temporal logic gate that differentiates between the start times of two chemical inputs and produces unique outputs accordingly (Fig 1A). The design relies on a system of two integrases with nested DNA attachment sites (Fig 1B). The use of integrases irreversibly inverts segments of DNA, resulting in a memory feature that can be maintained for multiple generations (Bonnet *et al*, 2012).

The design of the integrase temporal logic gate hinges on interleaving the attB attachment site of integrase B (intB) with the attP site of integrase A (intA), thus ensuring that the possible DNA flipping outcomes are mutually exclusive (Fig 1B). The serine integrases used in this design are TP901-1 (intA) and Bxb1 (intB). The fluorescent proteins mKate2-RFP (RFP) and superfolder-GFP

(GFP) are used both as placeholders for future downstream gene activation and as real-time readouts of the logic gate. The design also features a terminator (Bba-B0015) and a strong constitutive promoter (P7). In the case where there are no inputs, the terminator prevents expression of RFP from the constitutive promoter.

There are five possible basic events that could occur in a two-input system (Fig 1C): no input, inducer **a** only ($E_a$), inducer **b** only ($E_b$), inducer **a** followed by **b** at a later time ($E_{ab}$), and inducer **b** followed by **a** at a later time ($E_{ba}$). Consequently, in a perfectly resolved temporal logic gate, there should be five unique DNA states corresponding to the five types of events: $S_o$ (the initial state), $S_a$, $S_b$, $S_{ab}$, and $S_{ba}$. This design is limited to only four DNA states due to excision when $E_b$ occurs ($S_b = S_{ba}$). The two fluorescent outputs correspond to the two states that occur when inducer **a** is detected first—RFP is produced when the cell is in state $S_a$, and GFP is produced when the cell is in state $S_{ab}$.

Fig 1D illustrates the sequence of recombination that occurs during an event $E_{ab}$ that results in DNA state $S_{ab}$ and the production of GFP. Upon addition of inducer **a** at time $t_1$, TP901-1 flips the DNA between its attachment sites, reversing the directionality of the terminator and the Bxb1 attB recognition site (state $S_a$). Then, when inducer **b** is added at some time $t_2$ that is greater than $t_1$, the directionality of the Bxb1 sites is such that the DNA is flipped to reverse the directionality of the P7 constitutive promoter (state $S_{ab}$). If inducer **b** is added first (Fig 1E), the Bxb1 attachment sites are unidirectional—a configuration that results not in recombination, but in excision of the DNA between the sites (state $S_b$).

Once DNA recombination has occurred, it is irreversible. The unique attB and attP attachment sites are recombined into attL and attR sites, respectively, with which the integrases cannot bind to without additional cofactors (Ghosh *et al*, 2005). The nesting of the integrase attachment sites is the key design feature that produces the temporal **a** *then* **b** logic, and the irreversibility of the recombination records the event in DNA memory. The result is a genetic record that can both be sequenced later and immediately read via constitutive production of fluorescent outputs.

### A Markov model for integrase recombination

The most compelling advantage of engineered biological systems over man-made sensors lies in their inherent capabilities for replication and parallel sensing with minimal energy and resource requirements. Thus, deployment of synthetic bacterial devices would almost certainly involve populations of cells, never just a single cell. It is therefore important to understand how stochastic single-cell responses affect overall population-level distributions and outcomes. We created a Markov model of integrase-mediated DNA flipping and then used a stochastic simulation algorithm (Gillespie, 1977) to simulate individual cell trajectories (Fig 2A). All of the four possible DNA states are represented in the model: the original state ($S_o$), the intB excision state ($S_b$), the intA single flip state ($S_a$), and the **a** *then* **b** double flip state ($S_{ab}$). We have implemented the system experimentally by chromosomally integrating the target DNA into the genome of the *E. coli* cell. This allows us to assume that each cell only has one copy of the temporal logic gate (Haldimann & Wanner, 2001) and that each cell can be characterized

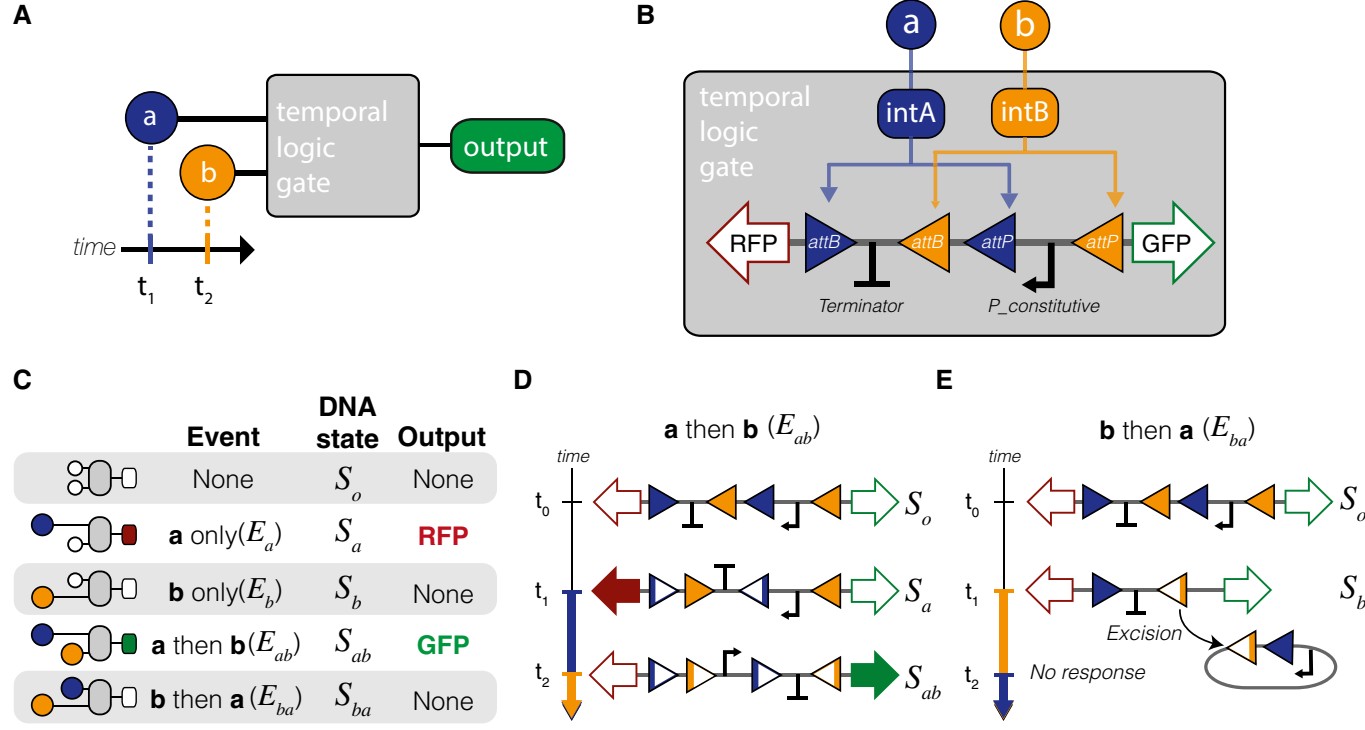

**Figure 1. Design overview of a temporal logic gate.**

A  A temporal logic gate distinguishes between two chemical inputs (**a**, **b**) with different start times.

B  Implementation of the temporal logic gate using a set of two integrases with overlapping attachment sites. Chemical inputs **a** and **b** activate production of integrases intA and intB, which act upon a chromosomal DNA cassette.

C  Table with all possible inputs and outcomes to the event detector.

D  Sequence of DNA flipping following inputs with inducer **a** before inducer **b** (event $E_{ab}$).

E  Sequence of DNA flipping following inducer inputs with **b** first (event $E_{ba}$). In any events in which **b** precedes **a**, the unidirectionality of the intB attachment sites results in excision.

by the tuple (DNA; IntA; IntB) (Fig 2B). The DNA terms are $S_o$, $S_a$, $S_b$, or $S_{ab}$, and IntA and IntB are non-negative integers representing the molecular copy number of each integrase. Once a DNA cassette has flipped into any of the states other than the original state $S_o$, there is no reverse process. The logic gate is designed such that if integrase B is expressed prior to integrase A, the DNA cassette is excised and the chain reaches the dead-end $S_b$ state. In order for a cell to successfully detect $E_{ab}$, it first needs to switch into state $S_a$ then transition into state $S_{ab}$ upon addition of inducer **b**.

Since each cell contains only a single copy of the temporal logic gate DNA, we can expect each cell to behave differently and to be highly susceptible to internal and external noise. This stochastic behavior will create a heterogeneous population response that can be analyzed for a more complex profile of event than if all the cells behaved uniformly. In order to capture the heterogeneity of cell population, we model the temporal logic gate using a stochastic model. Specifically, the stochastic transitions between the DNA states and the production/degradation of

integrases are mathematically modeled by a continuous-time Markov chain over the state space (DNA; IntA; IntB) as illustrated in Fig 2B. Definitions of transition rates can be found in Appendix Table S1.

*In silico*, the dynamics of a single cell translates to each stochastic simulation of the Markov model starting with (DNA = *So*; IntA = 0; IntB = 0) state. We define $\mathbb{P}_t(S_o); \mathbb{P}_t(S_a); \mathbb{P}_t(S_b)$; and $\mathbb{P}_t(S_{ab})$ as the probability that the DNA state of a single cell is $S_o$; $S_a$; $S_b$; and $S_{ab}$ at time $t$, respectively.

The temporal dynamics of the probability can be modeled by the following ordinary differential equation (ODE) (see equation 1). Where the notation $\mathbb{E}_t[\cdot|\cdot]$ stands for the conditional expected value at time $t$ (Full derivation, Appendix Section 12.1).

Serine integrases are produced as monomers that form dimers, search for specific attB and attP sequences, and, once both attB and attP sites are occupied, form a tetramer (dimer of dimers) that digests, flips, and re-ligates the DNA (Yuan *et al*, 2008; Rutherford *et al*, 2013). Though some cooperativity in ΦC31 binding to attB has

$$\frac{d}{dt}\begin{bmatrix} \mathbb{P}_t(S_o) \\ \mathbb{P}_t(S_a) \\ \mathbb{P}_t(S_b) \\ \mathbb{P}_t(S_{ab}) \end{bmatrix} = \begin{bmatrix} -\mathbb{E}_t[\alpha_1(\text{IntB})|S_o] - \mathbb{E}_t[\alpha_2(\text{IntA})|S_o] & 0 & 0 & 0 \\ \mathbb{E}_t[\alpha_2(\text{IntA})|S_o] & -\mathbb{E}_t[\alpha_3(\text{IntB})|S_a] & 0 & 0 \\ \mathbb{E}_t[\alpha_1(\text{IntB})|S_o] & 0 & 0 & 0 \\ 0 & \mathbb{E}_t[\alpha_3(\text{IntB})|S_a] & 0 & 0 \end{bmatrix}\begin{bmatrix} \mathbb{P}_t(S_o) \\ \mathbb{P}_t(S_a) \\ \mathbb{P}_t(S_b) \\ \mathbb{P}_t(S_{ab}) \end{bmatrix} \qquad (1)$$

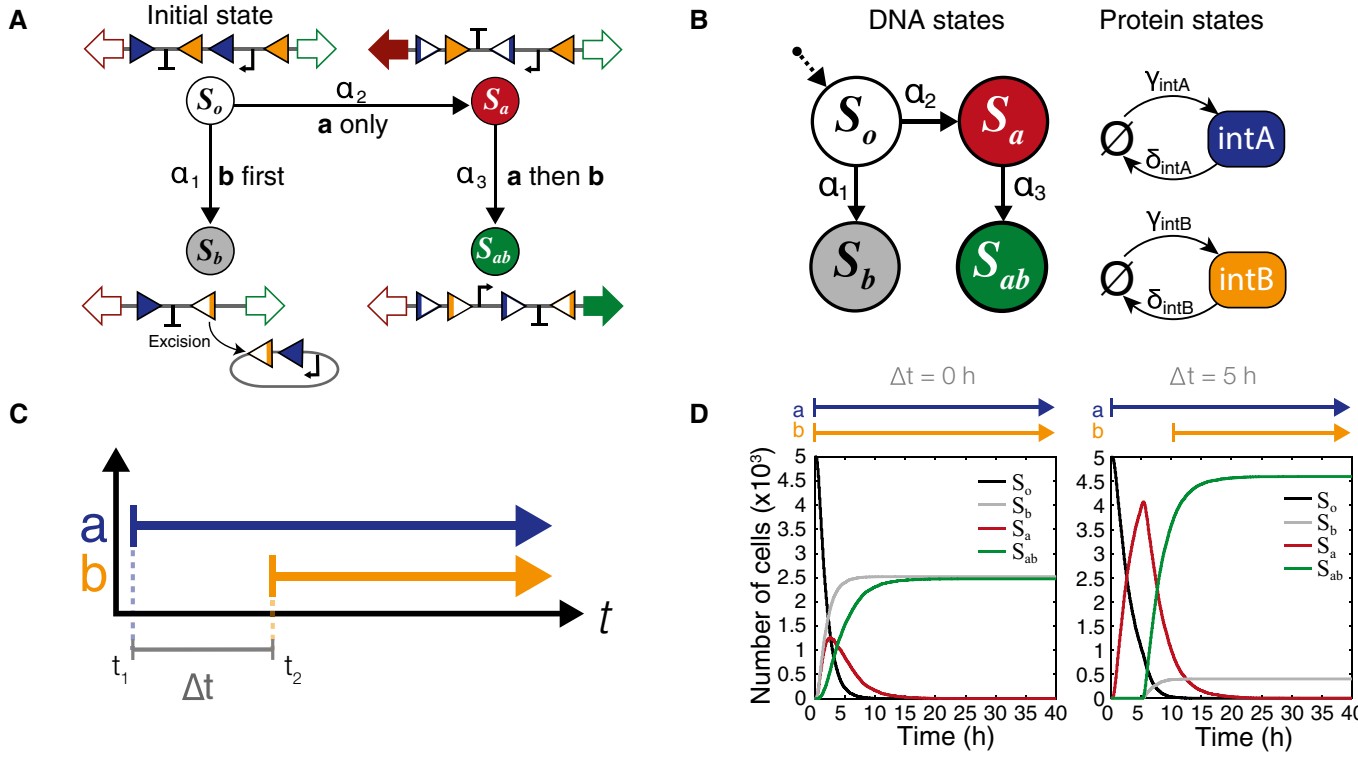

**Figure 2. A Markov model of integrase-mediated DNA flipping.**

A   The four possible DNA states, illustrated with DNA state diagrams. All DNA begins in the initial state $S_o$, and there are no reverse processes. The propensity functions $\alpha_1$, $\alpha_2$, and $\alpha_3$ are dependent on the concentration of the two integrases and correspond to the events **b** *first* ($E_b$), **a** *only* ($E_a$), and **a** *then* **b** ($E_{ab}$), respectively.

B   Representation of the same model as a Markov chain. Integrases are represented simply as protein states with production ($\gamma_A$; $\gamma_B$) and degradation ($\delta_A$; $\delta_B$) rates.

C   Graphical representation of inducer step functions. $\Delta t$ is defined as difference between the start time of the first inducer and start time of the second.

D   Simulation results for inducer separation times of 0 and 5 h. There are four possible DNA states, but all cells end up in either the $S_b$ or $S_{ab}$ final states. Individual trajectories are simulated for 5,000 cells and the number of cells in each DNA state is summed for each time point (Appendix Fig S2).

been found (McEwan *et al*, 2009), cooperativity in Bxb1 or TP901-1 integrase binding to attB and attP not been observed (Ghosh *et al*, 2005; Singh *et al*, 2014).

Rather than account for all individual DNA-integrase interactions, we have created a minimal model of stochastic transitions where only the final DNA states ($S_o$, $S_a$, $S_b$, $S_{ab}$) and the number of integrase monomer molecules (intA, intB) are tracked and all integrase activity is encompassed in the $k_{flip*}$ term. Since no cooperativity has been observed in Bxb1 or TP901-1 DNA binding (i.e., occupation of attB does not increase the probability of attP binding), we represent the required tetramerization as a fraction where flipping efficiency is zero unless at least four molecules are present. Thus, the propensity functions for state transitions as a function of integrase concentration, $\alpha_i(\mathrm{Int}_*)$, are defined in equation (2) where $\mathrm{Int}_*$ is integrase concentration; $K_{d*}$ is the dissociation constant; $k_{flip*}$ is the rate of flipping if the tetramer is formed; $i = 1; 2; 3$; and $* = \mathrm{A;B}$ (See Appendix Fig S1 for visualization of $\alpha_i(\mathrm{Int}_*)$ and Appendix Section 12.2 for full derivation).

We also define the time between the introduction of the first inducer ($t_1$) and the arrival of the second inducer ($t_2$) as the inducer separation time ($\Delta t$), such that

$$\Delta t = t_2 - t_1, \tag{3}$$

as shown in Fig 2C.

In the following set of simulations and experiments, we will consider cases with step inputs (Fig 2C), where the inducers are either present or not present. Concentrations of the inducers when they are "on" will be held constant. Also, it is important to note that inducer **a** is still present during and after time $\Delta t$ when inducer **b** is introduced.

Simulations of the Markov model were done with biologically plausible parameters in order to predict qualitative circuit behavior (Appendix Table S1). We limited the parameters to only the basic processes (integrase production, degradation, and DNA flipping), and parameter values were chosen to be within biological orders of magnitude. The single production rate constants, $k_{prodA}$ and $k_{prodB}$, combine the transcription and translation rates of each integrase.

$$\alpha_i(\mathrm{Int}_*) := k_{flip*}\left(\frac{\mathrm{Int}_*(\mathrm{Int}_* - 1)(\mathrm{Int}_* - 2)(\mathrm{Int}_* - 3)}{K_{d*}^4 + K_{d*}^3\mathrm{Int}_* + K_{d*}^2\mathrm{Int}_*(\mathrm{Int}_* - 1) + K_{d*}\mathrm{Int}_*(\mathrm{Int}_* - 1)(\mathrm{Int}_* - 2) + \mathrm{Int}_*(\mathrm{Int}_* - 1)(\mathrm{Int}_* - 2)(\mathrm{Int}_* - 3)}\right) \tag{2}$$

When an integrase in the model is induced, its production rate, $\gamma_*$, is the sum of $k_{prod*}$ and any leaky transcriptional expression, $k_{leak*}$ (*= intA or intB). The integrase monomer disassociation constant, $K_{d*}$, was estimated from measured Bxb1 binding constants (Singh *et al*, 2013). Parameter values for preliminary simulations were $k_{prodA} = k_{prodB} = 50(\mu m^3 \cdot h)^{-1}$, $k_{deg} = 0.3$ h$^{-1}$ (2.3 h half-life), $k_{flipA} = k_{flipB} = 0.4$ h$^{-1}$, $k_{leakA} = k_{leakB} = 0(\mu m^3 \cdot h)^{-1}$, and $K_{dA} = K_{dB} = 10$ molecules.

Our analysis of initial numerical simulation results highlights the significant role that the inducer separation time, $\Delta t$, plays in setting the final population distributions (Fig 2D). For each $\Delta t$, individual cell trajectories were generated with the assumption that each cell only has one copy of the target DNA ($N = 5,000$ trajectories). Then, at every time point, the total number of cells in each DNA state is counted (Appendix Fig S2). Fig 2D shows the contrast between adding both inducers simultaneously ($\Delta t = 0$ h) and adding inducer **b** after a 5-h delay ($\Delta t = 5$ h). Since both inducers are present by the end of simulation, all of the cells must have a final state that is either the $S_{ab}$ state or the $S_b$ state. No cells remain in the original $S_o$ configuration. $S_a$ is a transient state that builds up prior to the addition of inducer **b** and begins to convert to $S_{ab}$ immediately after the introduction of **b**. These initial simulation results suggest that $\Delta t$ may be a way to reliably tune the final population fractions of $S_{ab}$ versus $S_b$ state cells.

## Population distributions reflect inducer order and separation time

We used the model to further investigate the effects of varying both inducer order and separation time on population distributions in our experimental system and to understand the possible outcomes. In Fig 3, we simulate *in silico* cell populations that have been exposed to a sequence of overlapping step functions ($N = 5,000$ trajectories).

In the case of an $E_{ab}$ event, the proportion of cells that successfully detect *a* then *b* and switch to state $S_{ab}$ is a function of the inducer separation time, $\Delta t$ (Fig 3A). High $\Delta t$ means increasing the time that cells spend in only inducer *a*, allowing for most of the population to transition from $S_o \rightarrow S_a$ before the addition of any inducer **b**. Exposing cells to the inverse sequence of events, $E_{ba}$, results in a decrease of $S_{ab}$ cells proportional to increasing $\Delta t$ (Fig 3B). High $\Delta t$ in an $E_{ba}$ event means that $S_o \rightarrow S_b$ is the dominant reaction and cells that are partitioned into $S_b$ will not respond to inducer **a**. If we plot the final number of $S_{ab}$ cells from both $E_{ab}$ and $E_{ba}$ as a function of $\Delta t$ (Fig 3C), we see that the two curves do not overlap. $S_{ab}$ fractions exposed to $E_{ab}$ increase monotonically with $\Delta t$, while those exposed to $E_{ba}$ decrease monotonically with $\Delta t$. Thus, measuring the fraction of $S_{ab}$ cells by itself is sufficient to determine both the order of events and the timing, $\Delta t$, between them.

Additionally, we can define a detection limit, $\Delta t_{90}$, for which the inducer separation time results in $\geq 90\%$ of the population switching into the $S_{ab}$ state (Fig 3C). This $\Delta t_{90}$ limit provides a way to capture the two response regimes of the population. If the inducer separation time is less than the detection limit ($\Delta t < \Delta t_{90}$), then the rate of population switching is fast enough such that the number of $S_{ab}$ cells will correspond uniquely to some $\Delta t$ value. If $\Delta t > \Delta t_{90}$, then most cells have already switched to a final state,

and the differences in $S_{ab}$ cell count are too small to uniquely determine $\Delta t$.

The single-cell limitations of the temporal logic gate circuit can be overcome by measuring the number of $S_{ab}$ cells as a fraction of total cells. Though the logic gate itself does not have a unique genetic $S_{ba}$ state and cannot distinguish between a *b only* event versus a *b then a* event, these simulation results suggest that population-level fractional phenotypes can provide this additional information (Fig 3D). In the case of $E_{ab}$, fractions of $S_{ab}$ will always be above 50%, while $S_{ab}$ fractions less than 50% indicate $E_{ba}$. Additional figures showing how populations of $S_a$, $S_b$, and $S_o$ cells change with $\Delta t$ can be found in Appendix Figure S3.

*In vivo* step induction data supported model predictions and showed that population fractions of $S_{ab}$ cells could be tuned using $\Delta t$ (Fig 4). DH5α-Z1 cells were chromosomally integrated with one copy of the integrase target DNA and then transformed with a high-copy plasmid containing Ptet-Bxb1 and PBAD-TP901-1. When $\Delta t$ was varied from 0 to 8 h, we observed results qualitatively similar to model predictions. In Fig 4A, the cells have been exposed to an $E_{ab}$ event, where inducer **a** is present from time $t = 0$ h to $t_{end}$, and **b** is present from $t = \Delta t$ h to $t_{end}$. GFP expression during time course measurements is used as a proxy for $S_{ab}$ state cells, and flow cytometry was used to measure final populations. Comparisons of bulk fluorescence versus cytometry cell counts suggest that in single-copy integrants, overall GFP fluorescence is a good approximation of population $S_{ab}$ levels (Appendix Fig S12).

In Fig 4A, the number of cells in the GFP-expressing $S_{ab}$ state increases proportionally with increasing $\Delta t$ and continues to be responsive even when the two inducers are separated by 8 h. There is some expression of GFP in the presence of only inducer **a** ($E_a$), indicating some basal levels of intB. RFP expression, a proxy for the number of cells in state $S_a$, begins to increase at $t = 0$ h and drops at time $t = \Delta t$ when inducer **b** is added (Appendix Fig S4A). Aligning all of the GFP expression curves by $\Delta t$ (Appendix Fig S5) shows that lower values of $\Delta t$ not only have lower final GFP expression values, but also have slower rates of GFP production. This is consistent with modeling results because if we assume inducer **b** has an equal probability of entering any one cell, then in case of small $\Delta t$ ($\Delta t_{90} \leq 4$ h), there are a much larger number of $S_o$ cells and so the rate of $S_a \rightarrow S_{ab}$ state conversion will be lower. In the case of $\Delta t > 4$ h, the majority of cells in the population are already in the $S_a$ state configuration, and so the rate of cell state conversion to $S_{ab}$ will be much higher. When cells are exposed to $E_{ba}$, the number of $S_{ab}$ cells decreases monotonically with increasing $\Delta t$ (Fig 4B), and there is no RFP expression above background (Appendix Fig S4B). In both types of events, the cells maintained their state for up to 30 h in liquid culture and when re-streaked as single colonies. (Additional data with a more distinct color scheme and OD curves for this set of experiments can be found in Appendix Figs S6 and S7. Single-colony analysis in Appendix Fig S11.)

Final $S_{ab}$ (GFP) population fractions are sufficient to differentiate between populations that have been exposed to $E_{ab}$ versus $E_{ba}$ within 1 h of separation time between inducers (Fig 4C). Final populations after 30 h of growth were measured via flow cytometry and plotted against $\Delta t$. As $\Delta t$ increases, so does the $S_{ab}$ sub-population. The cells that encountered $E_{ba}$ have lower $S_{ab}$ fractions with high $\Delta t$, and at $\Delta t = 6$ h, the final $S_{ab}$ sub-population is equal

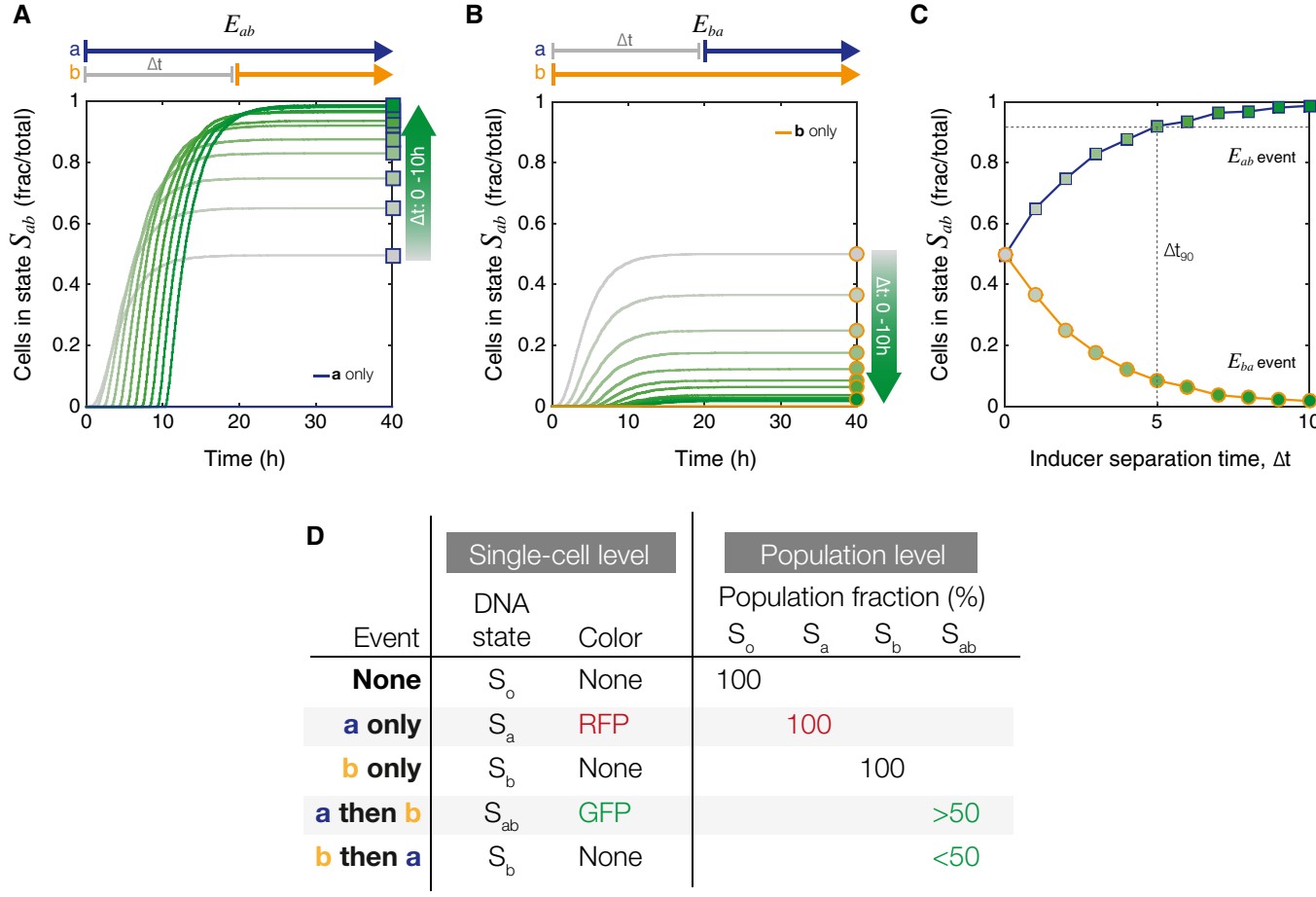

**Figure 3. Simulation results for inducer separation time for Δ*t* = 0–10 h.**

A   The population fraction (*N*/5,000 cells) that switches into state $S_{ab}$ following an $E_{ab}$ event is dependent on the inducer separation time, Δ*t*. The gray to dark green color gradient represents increasing Δ*t* values. Square markers indicate final population fractions for specific values of Δ*t*.

B   In the case of the inverse $E_{ba}$ event, the fraction of cells in state $S_{ab}$ decreases monotonically with increasing Δ*t*. Circular markers indicate final population fractions for specific values of Δ*t*.

C   Final $S_{ab}$ cell fractions from (A, B) are plotted as a function of Δ*t*. Blue line with square markers shows end point population fractions from an $E_{ab}$ event. Yellow line with circular markers shows final end point population fractions from an $E_{ba}$ event. The gradient inside the markers corresponds to increasing Δ*t* value. The dotted gray line corresponds to the Δ$t_{90}$, the value of Δ*t* at which ≥ 90% of the cells are in state $S_{ab}$. All simulations were done with a population of *N* = 5,000 cells.

D   Chart showing differences in information that can be recorded at the single-cell versus the population level. In particular, $E_{ba}$ does not have a unique single-cell genetic state, but has a clear distinct population-level phenotype.

to the baseline expression of the ***b* only** population, indicating that the addition of inducer **a** after a 6-h exposure to only inducer **b** has no effect at all. Based on where the GFP fraction exceeds 90% of the maximum $S_{ab}$ population fraction, the Δ$t_{90}$ detection limit for the experimental system is ~4 h. These experimental results show that the $S_{ab}$ population fraction clearly diverges for $E_{ab}$ and $E_{ba}$ when Δ*t* ≠ 0 h, indicating that $S_{ab}$ fractions alone can be used to determine both event order and separation time.

Further analysis of population-level data for all of the measurable fluorescent cell states can provide additional insights into differences in sub-population growth rates and leaky integrase expression (Fig EV1, Appendix Figs S8–S10). In Fig EV1, experimental populations from the step input experiments have been gated into quadrants such that $S_{ab}$, $S_a$, and $S_o + S_b$ populations can be counted. Even with maximum induction at highest Δ*t*, the maximum population fraction that can be switched appears to be approximately 60%

of the total population. We believe this is due to the non-fluorescent cells ($S_o$, $S_b$) having a slight growth advantage over differentiated cells. Studies have shown that unnecessary protein production has inverse effects on cell growth (Tan *et al*, 2009; Scott *et al*, 2010), and even with single-copy integrants, this would result in some overrepresentation of non-fluorescent sub-populations within the population. Single-colony analysis of the final populations shows that $S_o$ cells persist in the population even with 30–40 h of inducer exposure (Appendix Fig S11E).

Leaky expression of intA and intB can also be inferred from the *no inducer*, ***a* only**, and ***b* only** populations (Fig EV1A and B), and we can conclude that leaky expression is quite low, not exceeding ~0.5–3%. Even accounting for the overrepresentation of non-fluorescent cells, the baseline population split when both **a** and **b** are added simultaneously (Δ*t* = 0 h) is just under 50% of the total GFP population fraction. This suggests that the integrase flipping

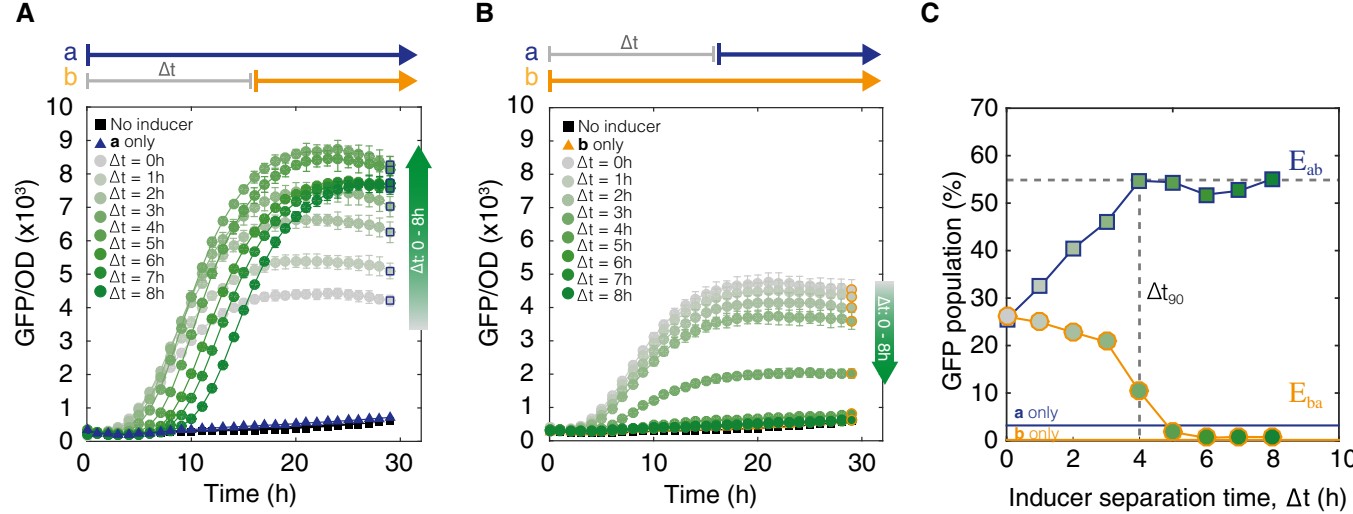

**Figure 4.** *In vivo* results for varying inducer separation time from $\Delta t$ = 0–8 h.

A   Populations of cells exposed to an $E_{ab}$ event sequence. Cell switching to state $S_{ab}$ (indicated by GFP fluorescence) begins when inducer **b** (aTc) is added. Maximum normalized GFP fluorescence increases as a function of the inducer separation time $\Delta t$. Gray to dark green gradient represents increasing $\Delta t$ values. Square markers are final end point measurements. Error bars represent standard error of the mean.

B   Cells exposed to the inverse $E_{ba}$ sequence of events. GFP fluorescence decreases monotonically with increasing inducer separation time between **b** and **a**. Circular markers are final end point measurements.

C   Final population distributions from (A, B) at 30 h are plotted as a function of $\Delta t$. Cells were gated by GFP fluorescence to identify percentage of $S_{ab}$ cells. Dotted line marks $\Delta t_{90}$ detection limit.

Source data are available online for this figure.

rates, $k_{flipA}$ and $k_{flipB}$, may not be equal and that the basal expression rates, $k_{leakA,B}$ should be nonzero.

## Varying model parameters for integrase activity and basal expression

Prior to proceeding with additional model-driven experimental designs, model parameters were modified to better represent asymmetrical integrase activity. The parameters for integrase flipping and leaky basal expression were tuned to account for the asymmetrical population responses to $E_{ab}$ versus $E_{ba}$ events (Fig 4C). We hypothesized that this asymmetry arises from a combination of unequal integrase activity when searching for and flipping the DNA, as well as leaky background expression of the integrases (Fig 5).

To understand overall trends in model behavior, we varied $k_{flipA}$ and $k_{leakB}$ while holding the other parameters constant. When the relative flipping efficiency of intA ($k_{flipA}$) was varied from 0.2 to 0.5 h$^{-1}$ ($k_{flipB}$ = 0.3 h$^{-1}$), we observed a bias in the baseline population split when both inducers are introduced simultaneously, $\Delta t$ = 0 h (Fig 5A, N = 3,000). Previously in the preliminary model (Fig 3C), the two integrases were assigned equal flipping rates, and the population split was expected to be 50/50 for $S_{ab}/S_b$. As the flipping rate of intA decreases relative to that of intB, that baseline shifts downwards to favor the more active integrase, intB. Varying the basal expression of intB ($k_{leakB}$) from 1% to 20% of the intB production rate ($k_{prodB}$) monotonically decreases the maximum $S_{ab}$ population fraction that can be reached in an $E_{ab}$ event (Fig 5B, N = 3,000). If there is a constant level of un-induced intB, then there will always be a minimum population of $S_b$ cells inhibiting the maximum fraction of $S_{ab}$ cells.

These simulation results showed that by varying $k_{flipA}$ and $k_{leakB}$, we could tune the baseline shift at $\Delta t$ = 0 and the maximum $S_{ab}$ ceiling at high $\Delta t$ to better approximate our experimental system. However, experimental measurements of leaky integrase expression showed that background expression was actually quite low (1% for intA, 1–3% for intB) (Figs 4C and EV1, **b** *only*, **a** *only*). Given actual measurements for $k_{leakA,B}$, we constrained those parameters and fit the model by varying $k_{flipA,B}$.

In order to find the best pair of values for $k_{flipA}$ and $k_{flipB}$, the flipping efficiency parameters for both integrases were varied from 0.1 to 0.6 h$^{-1}$ *in silico* (N = 500 cell trajectories), creating a matrix of simulated $S_{ab}$ population fractions for each combination (Appendix Fig S13). Leaky basal expression of the integrases was held constant based on experimentally measured values ($k_{leakA}$ = 1% of $k_{prodA}$, $k_{leakB}$ = 2% of $k_{prodB}$), and experimental data were normalized to a 70% population maximum for fitting purposes. Mean squared error was found by comparing model fits with experimental data (Appendix Fig S13A), and the combination with the minimum MSE was chosen (Appendix Fig S13B).

Fig 5C shows $\Delta t$ versus $S_{ab}$ population simulation results for final revised parameters. The final parameters were set to be $k_{flipA}$ = 0.2 h$^{-1}$, $k_{flipB}$ = 0.3 h$^{-1}$, $k_{leakA}$ = 0.01 · $k_{prodA}$(μm³ · h)$^{-1}$, and $k_{leakA}$ = 0.02 · $k_{prodB}$(μm³ · h)$^{-1}$ (Appendix Table S2). The introduction of leaky integrase expression into the model suggests that due to leaky expression of intB, around 2% of the population will "detect" $E_{ab}$ and be in state $S_{ab}$ even when no inducer **a** has been introduced. Additionally, preliminary simulation results suggest that the $\Delta t_{90}$ detection limit can be tuned by increasing or decreasing the overall production rate $k_{prod*}$ (*= A or B) (Appendix Fig S14), though this remains to be experimentally verified in future work.

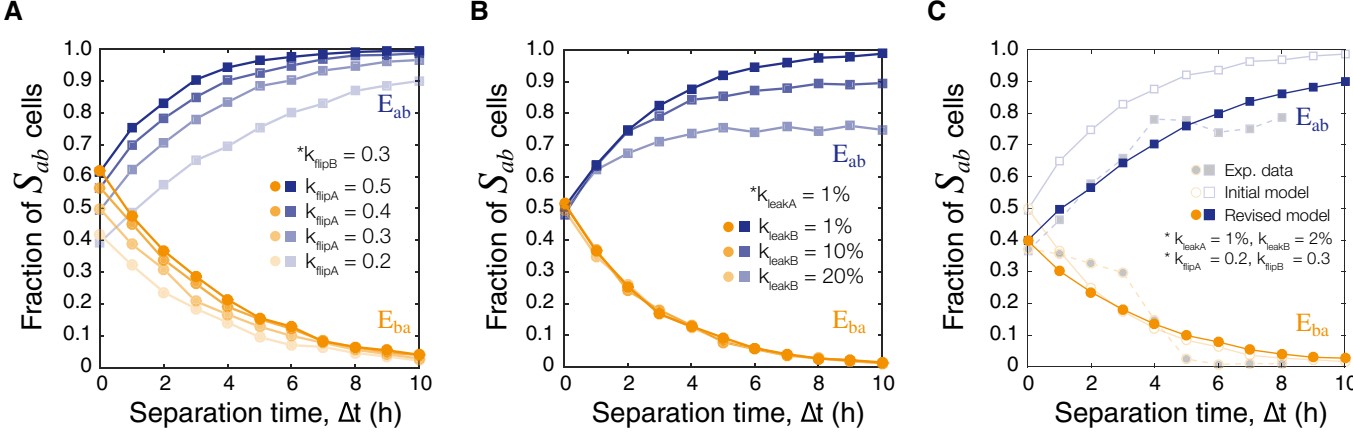

**Figure 5. Varying model parameters for integrase flipping and leaky expression.**

A   As DNA flipping rates of intA ($k_{flipA}$) are decreased relative to $k_{flipB}$, the population of $S_{ab}$ cells at $\Delta t = 0$ h has a downward shift. Simulations are done with $N = 3,000$ trajectories/marker.

B   Increasing the leaky expression of intB ($k_{leakB}$) changes the maximum threshold of cells that correctly identify $S_{ab}$ even at high $\Delta t$. Leakiness is defined as a percentage of the induced integrase production rate ($k_{prod*}$).

C   The model was revised to more closely match the experimental data by constraining parameters for leaky expression and varying integrase flipping ($N = 5,000$). Mean squared error was calculated between the experimental data and the initial and revised models to find an optimized pair of $k_{flipA,B}$ values (Appendix Fig S13). The revised parameters are $k_{flipA} = 0.2$ h$^{-1}$, $k_{flipB} = 0.3$ h$^{-1}$, $k_{leakA} = 1\%$ of $k_{prodA}(\mu m^3 \cdot h)^{-1}$, and $k_{leakB} = 2\%$ of $k_{prodB}(\mu m^3 \cdot h)^{-1}$.

*In silico* parameter space exploration shows that varying $k_{flip*}$ and $k_{leak*}$ parameters enables tuning of baseline $\Delta t = 0$ h split for $E_{ab}/E_{ba}$ and the maximum ceiling for $S_{ab}$ population fraction. Fold-change variations in relative rates allowed us to understand overall trends in the final populations, and we adjusted the model to account for inequalities in integrase flipping and leaky basal expression. Since leaky expression was measured to be small, we primarily tuned flipping rates. This process led us to more relevant model-informed predictions of experimental outcomes. With the refined model, we were interested to see whether distributions of the RFP-expressing $S_a$ state could provide information that measuring $S_{ab}$ fractions alone could not.

### Deducing pulse width from $S_a$ population fractions

Using the fraction of $S_{ab}$ (GFP) cells alone, we can determine $\Delta t$ values up to a $\Delta t_{90}$ limit for any given sequence of two step inputs. Now consider a pulse type of event, in which inducer **a** begins at time $t = 0$ h, remains constant throughout, and inducer **b** is introduced as a finite pulse at time $t = \Delta t$ h (Fig 6A). The start time of inducer **a** then becomes a reference for when the entire system is activated and ready to detect inducer **b**. Cell states are measured via flow cytometry at time $t_{end}$, where $t_{end} > 24$ h. Modeling results presented in this section are using the refined set of parameters defined in Fig 5C and Appendix Table S2.

If either of the two inducers is present in the media to some limit $t_{end}$, we would expect all of the $S_o$ cells to end up in one of two populations (Fig 6B). Cells that encounter inducer **b** first will be in the $S_b$ state, while cells that encounter **a** first will either be in the $S_a$ or $S_{ab}$ states. In the previous sections, once an inducer was added to the population, it was not removed, and the assumption was made that at times > 24 h, only a negligible number of $S_o$ cells remained. This type of step function induction also meant that only the final number of $S_{ab}$ cells (GFP) was needed to uniquely determine the

separation time $\Delta t$ because *any and all* cells that had switched to $S_a$ would eventually become $S_{ab}$.

However, in the case of a transient pulse, some cells that are in the $S_a$ state (RFP) will not ever encounter inducer **b**. Assuming that $k_{leakB}$ is small, these cells will remain in the $S_a$ state. Therefore, the population of **a** first cells equals $S_a + S_{ab}$. We simulated a matrix of populations exposed to varying inducer separation times ($\Delta t$) and inducer **b** pulse widths ($PW_b$) to measure the resolution of detectable events (Fig 6C). In simulation (Fig 6D), we can see that the two populations mirror each other to add up to 100% of the total cells ($N = 3,000$ cells, additional simulations in Appendix Fig S16).

Given that the step induction of **b** is equivalent to a pulse of infinite length ($PW_b = \infty$) and our prior experimental evidence showed that virtually no cells remain in state $S_a$ when $PW_b = \infty$, we reasoned that the final number of $S_a$ cells could be used to deduce information about the pulse width of **b**. This hypothesis was tested *in silico* by running a matrix of simulations with varying $\Delta t$ and $PW_b$. In Fig 6E, we see that the fraction of $S_a$ cells over the total number of cells decreases monotonically with increasing $PW_b$, and the curves overlap regardless of $\Delta t$. The overlap occurs despite nonzero leaky expression of intA and intB. The maximum number of $S_a$ cells does not go to 1 at $PW_b = 0$ h because of leaky intB expression ($k_{leakB} = 0.02\ k_{prodB}$).

Analytically, we solved equation (1) for $\mathbb{P}_t(S_a)$ to ensure that the $S_a$ population fraction is only dependent on $PW_b$. If inducer **a** is used as a constant reference signal, all cells transition into one of $S_a$; $S_b$; or $S_{ab}$ states, thus $\mathbb{P}_\infty(S_a) = 1 - (\mathbb{P}_\infty(S_b) + \mathbb{P}_\infty(S_{ab}))$. If we assume that the basal leaky expression of intB is zero ($k_{leakB} = 0$), $\mathbb{P}_t(S_b) + \mathbb{P}_t(S_{ab}) = 0$ holds for $t \leq \Delta t$, since there is no intB to switch the DNA state into $S_b$ or $S_{ab}$. Thus, we can show that $\mathbb{P}_t(S_b) + \mathbb{P}_t(S_{ab})$ is dependent only on $PW_b$, the duration of the pulse width of inducer B, for $t > \Delta t$. This conclusion holds as long as $k_{leakB}$ is negligibly small compared to other kinetic constants ($k_{flipA}$, $k_{flipB}$, $k_{deg}$, $\gamma_A$, and $k_{prodB}$) (See Appendix Section 12.3 for full derivation).

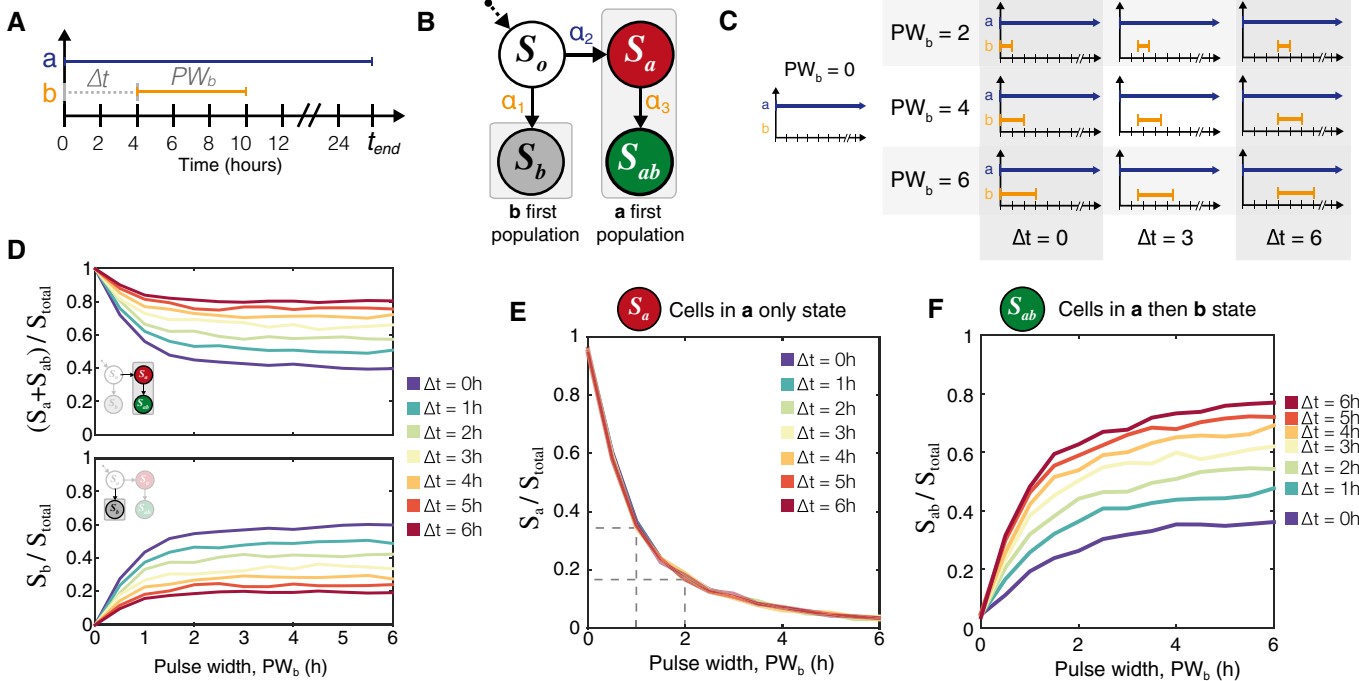

**Figure 6.  Simulation results for pulse width modulation.**

Simulations were done with revised parameters found in Fig 5C.

A   Inducer **a** can be used as a reference signal against which to measure the time and duration of the inducer **b** pulse.

B   The population eventually divides into one of two sub-populations: those that see inducer **a** first and those that see inducer **b** first. Only if a cell has entered the **a** *first* pathway does it have the possibility to express RFP or GFP. Furthermore, $S_a$ can be thought of as a necessary precursor to $S_{ab}$.

C   A matrix illustrating a subset of the $\Delta t$ and $PW_b$ values to be tested.

D   Simulation results show that for any given $\Delta t$, the number of cells in $S_b$ = (total number of cells − $(S_a + S_{ab})$)

E   The fraction of the population in the $S_a$ state is totally independent of $\Delta t$ and depends only on the pulse duration of inducer **b**.

F   Once $PW_b$ is known, then the fraction of the population in $S_{ab}$ state can be used to find the time at which the pulse of inducer **b** began. $N$ = 3,000 cell trajectories for each value of $\Delta t$, $PW_b$.

---

If $S_a$ population fractions can be modulated by changing $PW_b$, then conversely, we should be able to use measured experimental RFP population fractions as a way to determine $PW_b$. Once $PW_b$ is known, then the $S_{ab}$ fraction can be used to uniquely determine the time between inducers, $\Delta t$ (Fig 6F). Furthermore, the genetically encoded state means that these population fractions should be maintained and measurable at a time, $t_{end}$, that is much later than the time of the events.

These conclusions can be extended in simulation to create a scatterplot of $S_a$ cells versus $S_{ab}$ cells in a population (Fig 7A) over an 11 × 11 parameter matrix varying $\Delta t$ and $PW_b$ from 0 to 6 h in increments of 0.5 h (Additional plots in Appendix Fig S17). Each point on the chart in Fig 7A represents a simulated population ($N$ = 3,000) exposed to a unique combination of $\Delta t$ and $PW_b$ values. Vertical lines represent the same $PW_b$ value, and points with the same shape and color have the same $\Delta t$ value. The simulation results suggest sufficient resolution of events as long as $PW_b$ and $\Delta t$ values are between 0 and 4 h. For any single value of $PW_b$, we can follow the increasing $\Delta t$ values vertically and see that the population response saturates after 4.5 h resulting in overlapping between populations with 4.5 < $\Delta t$ < 6 h. We can trace any individual $\Delta t$ value horizontally from right to left and observe that the points begin to cluster and overlap when 4.5 < $PW_b$ < 6 h. These

simulation data suggest that there should be some defined detection range of $\Delta t$ and $PW_b$ where every possible combination of the two is uniquely identifiable.

Experimentally, we tested a 7 × 7 matrix of varying $\Delta t$ and $PW_b$ (0–6 h, 1 h increments) on independent populations of the temporal logic gate *E. coli* strain (Fig 7B). All populations, except for the control, were exposed to inducer **a** (L-ara 0.01%/vol) at time $t_0$ to $t_{end}$. Pulses of inducer **b** (aTc, 200 ng/ml) were achieved by sampling 5 μl of the population and diluting 1:100 into fresh media with only inducer **a** (M9CA + 0.01%/vol L-ara). Populations were collected and measured via flow cytometry after 24 additional hours of growth in inducer **a** (~36 h after start of experiment) (Fig EV2). For all values of $\Delta t$, the number of $S_a$ cells (RFP) is highest when there is no exposure to inducer **b** ($PW_b$ = 0 h) and decreases monotonically as a function of $PW_b$ (Fig 7B, top). We see a more pronounced separation of the $\Delta t$ curves when we look at $S_{ab}$ (GFP) cell fractions (Fig 7B, bottom). The number of $S_{ab}$ cells is dependent on both $\Delta t$ and $PW_b$ and increases proportionally with both increasing **b** pulse duration and inducer separation time.

By counting population fractions of RFP versus GFP-expressing cells, we can resolve the different populations that result from varying $\Delta t$ and $PW_b$ values (Fig 7C). As with Fig 7A, each point on the graph represents an independent population of cells (OD~0.7, ~10⁶

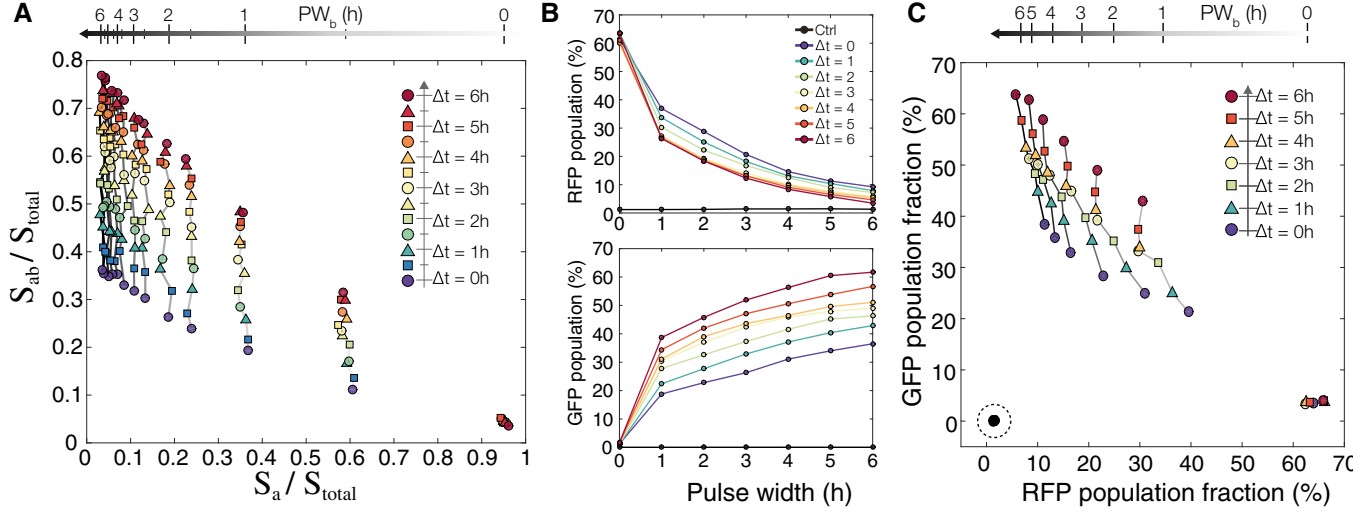

**Figure 7. Determining arrival time and pulse duration of inducer b with population fractions.**

A  Simulation results from testing an 11 × 11 matrix of parameters with $\Delta t$ and $PW_b$ varying from 0 to 6 h in increments of 0.5 h. Each point represents a population of 3,000 cells. Increasing $PW_b$ goes from right to left, and increasing $\Delta t$ goes from bottom to top.

B  Experimental results showing RFP and GFP population fractions as a function of increasing $\Delta t$ and $PW_b$. Experimental results were obtained by exposing temporal logic gate *E. coli* populations to varying $PW_b$ and $\Delta t$ values (0–6 h, 0.01%/vol L-ara, 200 ng/ml aTc, measurements taken at 48 h).

C  A scatterplot of each population using their RFP and GFP fractions as coordinates (~$10^6$ cells per population). The non-induced control samples are indicated with a dotted circle on the bottom left, and the samples with $PW_b = 0$ h are on the bottom right. Samples with the same $PW_b$ are connected with a solid line, and line darkness represents increasing $PW_b$ duration. Samples with the same $\Delta t$ are shown with the same colored shape marker and increasing $\Delta t$ goes from bottom to top.

Source data are available online for this figure.

cells counted per population). All of the populations exposed to either or both of the inducers occupy fractional coordinates that are unique from that of the *no inducer* controls (indicated by dotted circle). We see that if $\Delta t$ is constant and $PW_b$ increases (Fig 7C, right to left), then the $S_a$ fraction decreases as $S_{ab}$ fractions increase. For constant $PW_b$ with increasing $\Delta t$ (Fig 7C, bottom to top), the $S_a$ cell fraction remains mostly constant relative to increasing $S_{ab}$. In the case where there is no **b** pulse ($PW_b = 0$ h), we see maximum $S_a$ (RFP) cell fractions of about 60% with minimal $S_{ab}$ populations that are about the same as *no inducer* $S_{ab}$ levels. Overall, populations with different $PW_b$ exposures are well separated by $S_a$ (RFP) fraction up to 4 h. Even for $PW_b$ at 5 and 6 h, the populations have unique $S_a/S_{ab}$ coordinates, just not unique $S_a$ fractional values.

This method of profiling is only valid if the fraction of $S_a$ state cells can be used as a measure of $PW_b$ that is independent of $\Delta t$. In previous experiments with step inputs (Fig EV1), there would be a significant population of cells with both GFP and RFP fluorescence, since they had transitioned to $S_{ab}$ but had not yet fully diluted out built up RFP protein levels from being in $S_a$ for extended periods. If a significant percentage of the population remained in this transition state (Q2), that would make RFP an unreliable measure of $S_a$ state cells. However, flow cytometry analysis of the pulse-modulated populations (Fig EV2) showed that although there were some cells expressing both RFP and GFP (Q2), these cells were always < 3% of the total population. (Additional flow cytometry analysis can be found in Appendix Figs S18–S23.) Thus, RFP was measured to be a reliable determinant of $S_a$ state cells, and subsequently, of $PW_b$.

For any given $PW_b$, we observed higher experimental $S_a$ (RFP) population fractions with lower $\Delta t$ (Fig 7A top), resulting in a diagonal slant for each value of $PW_b$ (Fig 7C). Upon further investigation, we believe this is due to a slower $S_o \xrightarrow{\alpha_1} S_b$ transition than we anticipated. In our model, we assume $S_o \xrightarrow{\alpha_1} S_b$ is equal to $S_a \xrightarrow{\alpha_3} S_{ab}$, since both transitions are mediated by intB. However, the gradual decrease in $S_a$ fractions with increasing $\Delta t$ for each value of $PW_b$ suggests that the $\alpha_1$ transition rate may be actually be slower than $\alpha_2$ or $\alpha_3$. Simulation results with adjusted transition rates ($\alpha_1 < \alpha_2 = \alpha_3$) recapitulated the slanting $S_a$ population fractions (Appendix Fig S25). This inequality in transition rates could have arisen from differences in DNA sequence length or from differences in the DNA excision required for $S_o$ transition to $S_a$ instead of the recombination that occurs in the other transitions. Differences in DNA excision or recombination for a single integrase are important experimental parameters, but do not ultimately affect our conclusions about the overall system. Despite unequal intB transition rates, experimental implementation of the temporal logic gate still produces unique ($S_a$, $S_{ab}$) fractional coordinates for each combination of $\Delta t$, $PW_b$, even though $S_a$ values are not unique for higher $PW_b$.

Model-informed predictions on population fractions in response to pulses of inducer **b** led to experiments that could produce unique $S_a$ and $S_{ab}$ coordinates for different combinations of $\Delta t$ and $PW_b$. However, experimental data also revealed areas in which the model had been oversimplified. While it is important to have a model to understand overall properties and limitations of the experimental system, it is also impractical to design simulations that can account for all possible variations that might occur in the implementation of

                    

biological devices. Therefore, we believe that future workflows should also involve calibration protocols for specific applications of engineered biological populations.

### Practical use and calibration of populations for event detection

Curve-fitting methods were used to automatically convert experimentally measured RFP and GFP population fractions into $PW_b$ and $\Delta t$ values and to evaluate the resolution with which population ratios can be used to determine inducer separation time and pulse duration. Using the experimental data from Fig 7B and C, we generated fitting curves for PWb as a function of RFP population percentage ($R$) and for $\Delta t$ as a function of both GFP population percentage ($G$) and $PW_b$ (Appendix Figs S26 and S27, Appendix Table S3). We will denote these functions with $PW_b(R)$ and $\Delta t(G, PW_b)$, respectively. The functions $PW_b(R)$ and $\Delta t(G, PW_b)$ can then be used to generate a mesh of estimated $PW_b$ and $\Delta t$ values for any given normalized fluorescence values (Fig 8A, Appendix equations 8–11).

The estimated values were compared against the actual values to determine the approximate time window with which a specific $PW_b$ or $\Delta t$ value can be resolved. For each actual value of $PW_b$ and $\Delta t$, we calculated the average and standard deviation for the set of estimated values. The standard deviation allows us to visualize the range for which the majority of predictions will fall for any given actual value. For instance, a $PW_b$ of 1 h can be detected $\pm$ 0.25 h, but as $PW_b$ increases, this prediction window widens and for $PW_b \geq$ 3 h, the resolution of detection is closer to $\pm$ 1 h (Fig 8B). Similarly, predicted values of $\Delta t$ fall within $\pm$ 0.5 h for $0 < \Delta t < 3$ h and increase to $\pm$ 1 h when $\Delta t \geq 3$ h (Fig 8C). Using these fitting functions, we can also pre-generate a reference table that converts RFP and GFP population fractions into predicted $PW_b$ and $\Delta t$ values (Appendix Table S4).

## Discussion

Engineered biological systems have inherent capabilities for replication, parallel processing, and energy efficiency. These advantages rely on the existence of bacteria not as single cells, but as populations. As the field moves forward with synthetic gene circuits, it is important to understand outcomes not just as single-cell outputs but as overall population-level distributions.

We have designed and implemented a temporal logic gate that takes advantage of the population dynamics to collectively sense and record sequences of transient chemical inputs. We show both that single cells independently sense and record events and that aggregate population fractions create unique outcomes that provide information not encoded in single cells. As with all engineered systems, proper calibration of these temporal logic gate populations will be required prior to deployment in the "field". We envision a process similar to the one described in this report. First, experimental populations are exposed to a matrix of $PW_b$ and $\Delta t$ values. This will set the maximum and minimum RFP and GFP population fractions and provide necessary data for determining the $\Delta t_{90}$ limit and producing the fitting functions $PW_b(R)$ and $\Delta t(G, PW_b)$. Once the fitting functions have been determined, values for $PW_b$ and $\Delta t$ for experimental samples can be estimated within $\pm$ 0.25 to 1 h of the actual values. A calibrated table could also be generated and used for as a reference for samples that have been exposed to unknown conditions.

The stochastic nature of molecular processes often presents a significant barrier to homogenous outputs from an engineered population of cells. This implementation of event detection via population fractions takes advantage of stochastic and heterogeneous individual responses to environmental conditions in order to map final population fractions back to unique sequences and durations of chemical events. The sensitivity of the system and the $\Delta t_{90}$

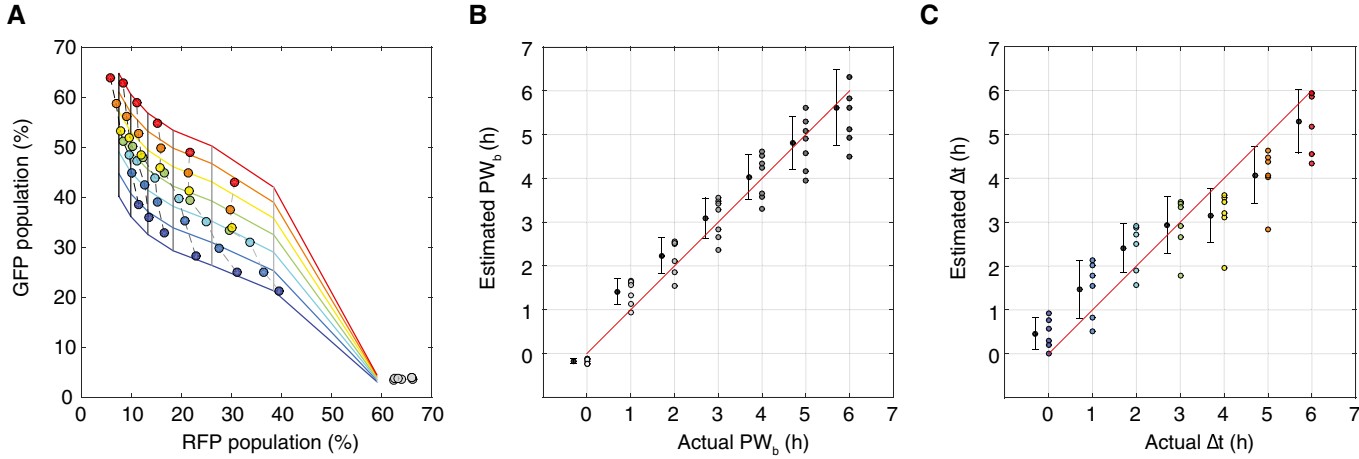

**Figure 8.  Determining prediction resolution for $PW_b$ and $\Delta t$ from population data.**

A   A mesh generated from fitted curves for $PW_b$ as a function of RFP population percentage($R$) and $\Delta t$ as a function of pulse width and GFP population percentage($G$). Experimental data are overlaid.

B   Comparison of actual versus estimated $PW_b$ values generated by fitted function $PW_b(R)$. For each actual $PW_b$ value, the average of the estimated $PW_b$ values with $\pm$ 1 standard deviation (slightly offset on the *x*-axis for better comparison).

C   Comparison of actual versus estimated $\Delta t$ generated by the fitted function $\Delta t(G; PW_b)$. For each actual $\Delta t$ value, the average of the estimated $\Delta t$ with $\pm$ 1 standard deviation (slightly offset on the *x*-axis for better comparison).

Source data are available online for this figure.

detection limit could potentially be modulated by increasing or decreasing protein production rates via tuning of plasmid copy numbers, signal concentration, or transcription/translation sequences. The use of digital cellular outputs combined with the analog population response creates event detection systems that are more robust to stochasticity and can be tuned more easily. We plan to explore these possibilities in future work.

As a proof-of-concept, we have used the common laboratory inducers L-arabinose and aTc as inputs, but we anticipate that our temporal logic gate system could be modularly adapted to any pair of biosensors. In particular, we believe there are possibilities for detection of miRNAs and biofilm formation. Stable populations of microRNAs (miRNAs) circulating in the blood have generated a lot of interest as biomarkers for human health (Cortez *et al*, 2011). These short (~20–30nt) regulatory RNAs have been shown to have sequential tissue-specific expression signatures that correlate with pregnancy, tumor formation, and other diseases (Gilad *et al*, 2008; Mitchell *et al*, 2008), and synthetic biology has developed many customizable RNA sensors (Friedland *et al*, 2009; Green *et al*, 2014). Detection of miRNAs would require implementation of the temporal logic gate in mammalian cells. Though recombinase-based synthetic circuits have not been shown in mammalian cells, serine integrases have been used quite effectively in a wide variety of mammalian cell types, primarily for genome editing and integration (Keravala *et al*, 2006; Xu *et al*, 2013).

Another possible application would involve the detection of harmful biofilms. Biofilms are self-assembling, highly structured, multi-species consortia that develop in stages and have sophisticated networks of interaction and function (Stoodley *et al*, 2002; Flemming & Wingender, 2010; Elias & Banin, 2012). Unnatural biofilm development in environments such as industrial water sources or waste streams can be both harmful for both the natural environment and the industrial mechanisms. Detection of biomarkers for known strains of biofilm colonizers would provide early warning of changing ecosystems, and although we do not yet fully understand these networks, it is known that quorum sensing plays a critical role in the process. Quorum-sensing molecules and receptors are available in the synthetic biology toolbox and so may provide an accessible way of detecting the sequential colonization of different microbes. Field deployment of engineered bacteria will likely involve transient signals, low-nutrient environments, and possibly even other microbial competitors (i.e., soil, flowing rivers, the digestive tract). We used minimal media in this study to better approximate low-nutrient environments, and anticipate further characterization in more customized "local" environments (i.e., gut model or air model or soil model) and with hardier microbial chassis.

Finally, this study focused on the population outputs as indicators of past events, but we believe that this temporal logic gate could be used to reliably differentiate a single strain into controlled sub-populations via input pulse order, duration, and frequency. In recent years, it has been recognized that many natural systems modulate cellular behavior not only by changing the concentration of signaling molecules but also by regulating signal pulse frequency (Cai *et al*, 2008; Lin *et al*, 2015). If we consider the fluorescent proteins GFP and RFP in this circuit as simply placeholders for downstream genes, then this system could easily be applied as a top-down population differentiator. By modulating the sequence of inputs, one could systematically predict and create mixed populations of genetically differentiated cells. This greatly expands our capability to design synthetic systems that have controllable distributions as outcomes, not just digital on/off phenotypes. Furthermore, we can then begin to develop frameworks for understanding the role of feedback and control theory in modulating these sub-populations given different starting distributions or uneven growth rates due to resource limitations. As the scientific community turns toward further understanding of microbiomes and multicellular consortia, engineered bacteria populations could be used not only as a tool for investigating the activities of natural communities but also as a way to build synthetic communities from the ground up.

# Materials and Methods

### Cell strains and plasmids

All plasmids used in this study were designed in Geneious 7.1 (Biomatters, Ltd.) and made using standard Gibson isothermal cloning techniques. Integrases Bxb1 and TP901-1 are on a high-copy plasmid (pVHed05, plasmid map in Appendix Fig S9) with a ColE1 origin of replication (original template from the dual recombinase controller (Bonnet *et al*, 2013), Addgene Plasmid 44456). Integrase A (Bxb1) is behind a Ptet promoter and integrase B (TP901-1) is behind a PBAD promoter. The plasmid has been modified with an additional TetR gene. The temporal logic gate was integrated into the Phi80 site on the *E. coli* chromosome using CRIM integration (Haldimann & Wanner, 2001) and screened for single integrant colonies. The integration plasmid template and DH5α-Z1 strain were generously provided by J. Bonnet and D. Endy and modified to contain the temporal logic gate (pVHed07, plasmid map in Appendix Fig S9). Additional DNA and oligonucleotides primers were ordered from Integrated DNA Technologies (IDT, Coralville, Iowa). A custom formulation of M9CA media was used for all experiments. The media contained 1× M9 salts (Teknova, M1906) augmented with 100 mM $NH_4CL$, 2 mM $MGSO_4$, 0.01% casamino acids, 0.15 μg/ml biotin, 1.5 μM thiamine, and 0.2% glycerol and then sterile-filtered (0.2 μm).

### Model simulations

The stochastic simulation algorithm by Gillespie (Gillespie, 1977) was implemented to generate the sample paths of individual cells using the Markov model (see Appendix Table S6 for the definitions of Markov transitions and transition rates). All simulation runs and their analyses were done with MATLAB (R2014b, The MathWorks, Inc.). Simulated populations were done with 3,000–5,000 individual cell trajectories. Source code for MATLAB simulations is available as Code EV1.

### Experimental methods

Prior to all experiments, cells were grown overnight from plate cultures in M9CA for 2 days, then diluted to OD 0.1 and recovered for 4–6 h at 37°C. L-arabinose and anhydrous tetracycline (aTc) were used as inducers **a** and **b**, respectively. L-ara was used a concentration of 0.01% by volume, and aTc was used a

concentration of 200 ng/ml (450 nM). All media contained the antibiotics chloramphenicol (Sigma-Aldrich, Inc (C0378); 50 μg/ml) and kanamycin (Sigma-Aldrich, Inc (K1876); 30 μg/ml). All experiments were performed with the aid of timed liquid handling by a Hamilton STARlet Liquid Handling Robot (Hamilton Company).

For step function experiments, the cells were diluted to OD 0.06–0.1 into a 96-well matriplate (Brooks Automation, Inc., MGB096-1-2-LG-L) with 500 μl total volume in M9CA. Cultures were incubated at 37°C in a BioTek Synergy H1F plate reader with linear shaking (1096 cycles per minute) (BioTek Instruments, Inc.), and inducers were added at appropriate time by the Hamilton robot. OD and fluorescence measurements (superfolder-GFP ex488/em520, mKate2-RFP ex580/em610) were taken by the BioTek every 10 min. Each experimental condition was done on the plate in triplicate.

For the pulse experiments, single 500-μl cultures were grown at 37°C in the BioTek plate reader (linear shaking, 1096 cycles per minute) and inducers added at time Δ*t* by the Hamilton liquid handler. Pulses were achieved through dilution of the culture into fresh M9CA media containing 0.01% L-arabinose. The Hamilton was programmed to sample 5 μl of the culture and dilute it into 500 μl of fresh M9CA + 0.01% L-ara to achieve pulsatile exposure to aTc. This was done in three independent triplicates for each experimental condition. About 96-well deep-well plates containing the diluted cultures were then incubated at 37°C incubated for an additional 24 h (~36–40 h from start of experiment). Final end point populations were measured using the plate reader and also stored and further analyzed using flow cytometry.

Analysis of experimental data was done using custom MATLAB scripts. All depicted error bars are standard error of the mean. Fitting of curves was done in MATLAB.

### Flow cytometry

Experimental cultures were spun down, washed, resuspended in sterile PBS with 15% glycerol, and stored at −80°C (Jahn *et al*, 2013). Cultures were then thawed on ice and diluted to $10^6$ cells/ml in sterile PBS prior to running on the flow cytometer. Flow cytometry was done using a MACSQuant VYB (Miltenyi Biotec, Germany) at the Caltech flow cytometry core facility. Flow data analysis and gating was done with FlowJo version 10.0.8r1 (Flowjo, LLC, Ashland, OR). For inducer separation time experiments shown in Figs 4 and EV1, ~$10^5$ cells were measured per population. For pulse induction experiments shown in Figs 7 and EV2, ~$10^6$ cells were measured per population.

**Expanded View** for this article is available online.

### Acknowledgements

The authors would like to thank J. Bonnet and D. Endy for the initial plasmids used in this work, S. Sanchez for critical assistance in automation and liquid handling, D. Perez for flow cytometry assistance, and C. Hayes for discussions. V.H. is supported by the U.S. Department of Defense (DoD) through the National Defense Science & Engineering Graduate Fellowship (NDSEG) Program. Y.H. is supported by JSPS Fellowship for Research Abroad. Research supported in part by the Institute for Collaborative Biotechnologies through grant W911NF-09-0001 from the U.S. Army Research Office. The content of the information does not necessarily reflect the position or the policy of the Government, and no official endorsement should be inferred.

### Author contributions

VH conceived of the circuit design, constructed the necessary experimental strains, performed experimental work and data analysis, ran model simulations, and wrote the manuscript. YH developed the stochastic model and derived the mathematical results. PWKR provided feedback and guidance on data analysis and interpretation. RMM provided feedback and guidance on overall project vision, circuit design, and interpretation of results.

### Conflict of interest

The authors declare that they have no conflict of interest.

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
