## [Review Process File · Molecular Systems Biology]

A population-based temporal logic gate for timing and recording chemical events

Victoria Hsiao, Yutaka Hori, Paul WK Rothmund, Richard M Murray

Corresponding author: Victoria Hsiao, California Institute of Technology

Review timeline:

Submission date:	27 October 2015
Editorial Decision:	10 January 2016
Revision received:	06 April 2016
Accepted:	10 April 2016

Editor: Thomas Lemberger

Transaction Report:

1st Editorial Decision

10 January 2016

Thank you again for submitting your work to Molecular Systems Biology. First of all, I would like to greatly apologize for the lengthy process, which was due to the late arrival of the reports and the Christmas holiday. We have now finally heard back from the three referees who agreed to evaluate your manuscript. As you will see from the reports below, the referees find the topic of your study of potential interest. They raise, however, several issues that should be carefully addressed in a major revision of the present work.

Without repeating all the points made by the reviewers in their reports, one of the major issue raised by the reviewers and that needs to be addressed is to complement the analysis by performing single-cell measurements (cytometry) to assess directly whether the heterogeneity of the cell populations is as predicted.

We would highly recommend to make the text as accessible as possible to a general audience. If necessary, you can include one or two "Boxes" to provide some didactic explanation on your approach and the modelling. This would make your work potentially more approachable to general readers.

REFEREE REPORTS

Reviewer #1:

The authors set out to show that the DNA integrase circuit presented in the manuscript can be used to deduce the duration of a chemical pulse and the timing between two chemical events. They also develop a model to explain the experimental observations of their circuit as well as to suggest different ways it can be used to gather information about two inputs. This is an interesting paper in concept, but the experimental methods are inadequate for validating the model, which I have several questions about as well. Thus, I believe the paper would need to be significantly improved before it should be considered for publication. Below are a list of comments and suggestions for the authors:

1. Authors model the rate of integrase-mediated flipping as being linearly proportional to the amount of integrase, but this is an over-simplification. The authors themselves acknowledge that integrase-mediate recombination requires multimerization of the integrase (page 6, second to last paragraph). and so one would expect that recombination rate would be non-linear with respect to integrase concentration. Their simplification appears to be without justification.
2. Authors use a degradation constant of .01/hr for the integrases in their model and say that it corresponds to a ~69 minute half life (page 6 main text, page 12 supplement), but this actually corresponds to a ~69 hour half life, which seems unrealistic.
3. Authors use population-level fluorescence measurements in their experiments and therefore never directly assay the heterogeneity of their cell populations to see if they are dividing into different states as expected. To better validate that their circuit is performing as expected, I recommend that the authors use cytometry for high-throughput single-cell measurements instead of population-level measurements so that they can directly measure the fractional size of different fluorescent subpopulations. For example, the authors claim that integrating the DNA target sites for the integrases into the genome results in a single-copy per cell, but this may not be true when cells are actively replicating. How does this affect the behavior of their circuit at the single cell level?
4. The authors should give more information about the growth of the cells during their experiments. Presumably the cells switch from exponential to stationary phase at some point during their experimental time courses. For example, if the second inducer is added closer or further away from stationary phase will it affect the way that the population divides into different states?
5. Overall, more information in the experimental section would be useful. For example, how many replicates did the authors use in their experiments?
6. With regard to the model fitting (figure 5C), authors should perform a more thorough analysis of the estimated parameters from the fit. For example, are these

parameters a unique solution or is it possible that a completely different set of parameters could have given just as good of a fit? If the latter is the case, then the estimated set of parameters could be meaningless.

7. Authors claim that the Δt_{90} detection limit can be tuned by modulating the overall production rate of the integrases. The authors should show this experimentally. It seems like production rates could be tuned by mutating the ribosome binding sites of the integrases or the promoters from which they are expressed.

8. One page 12 paragraph 2 the authors say "Our numerical simulations predicted a complete overlap of the Sa curves (figure 6E), and the experimental results are consistent with those predictions, though there is some downwards drift with increasing Δt ". This sentence contradicts itself. It suggests that fraction of cells in Sa is invariant to Δt even when the data (figure 7B) clearly shows a trend with Δt . The authors need to be more upfront about this incongruence between their model and the experimental data and suggest reasons why it might be happening and how it might be resolved. For example, is it an integrase leakiness issue?

9. In the experimental methods section, the authors forgot to put the source and concentration of chloramphenicol and kanamycin- instead it just reads "source and concentration".

10. In the discussion, the authors talk about how this system could be adapted to microRNA detection. It is unclear how this would work in the context of the current system - would the microRNAs be taken up into the bacteria, and if so, how? Or are the authors alluding to some in vitro system?

11. First sentence of paragraph 2 on page 3, the word "chemical" is written twice in a row.

12. In the final paragraph of the introduction "E. coli" is not italicized.

13. On page 5 the statement that the attL and attR sites are "no longer recognized by the integrase" warrants a citation.

14. Authors claim that GFP and RFP provide "real-time" readout of the temporal logic gate, but they do not take into account the time delay it takes to express these proteins to appreciable levels for measurement.

15. Why does GFP fold change peak over the experimental time course in Figure 4A?

16. Is fluorescence being normalized by OD? What is the relative growth rate of cells exposed to different combinations of the inputs, and does this affect the measurements? It would be helpful to show the OD of the different curves and explicitly state how data is being processed in the methods / captions.

17. Do the Eba measurements (Fig 4B and 4C) not have error bars, or are they just really small?

18. Color choices for figure 4A and 4B make it hard to distinguish lines belonging to different Δt values.

19. In Figure 6E, all of the lines are hidden behind the red-line because of overlap, is there a way to avoid this?

20. In Figure S1, I do not understand the plots showing the trajectories of cell state (S_o , S_a , S_b , S_{ab}). Why are these plots labeled "copies per cell" on the y-axis and why do trajectories sometimes have values that are not Boolean over time? If the lines represent whether or not a cell is in a state, shouldn't they just be jumping between two Boolean values ("in state" versus "not in state")?

Reviewer #2:

The authors demonstrate, both computationally *in silico* and experimentally *in vivo*, an integrase-based logic gate design for genetic switching that enables cells to discern molecular information at a population level. Here, cell populations discriminate between the order of two events (exposure to two analytes/inducers) and also reveal their timing and duration. In this particular example, the genetic switching associated with the events 'a' and 'b' also provide a DNA-based detailed record (where the presence of an inversion documents event order and its frequency is associated with timing); this acts as permanent memory of the exposure to a and b. These results, in combination with critical model optimization and calibration for interpretation of results, provide rich information about the specific consequences of many induction factors (lapse between events, duration of events, stochastic nature) on the output population state.

Further, the work provides valuable insight to the synthetic biology community about how the perceived challenge and limitation of stochastic noise associated with genetic regulation can, in fact, be exploited to tease out precise event details that influence a particular outcome distribution.

However, the introduction of this work under emphasized the impact of population-level analysis compared to the single cell contribution. Based on the presented results, a population can achieve behavioral response that a single cell cannot-specifically, the genetic circuit design does not permit detection of the event E_{ba} , yet the population as a whole produces a titratable response due to stochastic noise. This concept of extending design considerations beyond a single cell and its individual engineer-able feedback regulation is novel, and so more elaboration on this important point is essential.

Overall, though, this work provides an impressive pair of simulated and real results that show predictable and informative population behavior, which is compelling toward expanding synthetic biology strategies.

Major Comments:

-More references on the topic of utilizing cell populations for classification of chemical/molecular input would be helpful. For example there is work on cell networks for conveying molecular communication that is highly relevant to this work.

-The hierarchical organization of the system, and its contributions to the output at each level, is a compelling feature. Thus, it would be helpful to strongly differentiate between the respective roles of the single cell (logic design, model of cell response parameters, etc) and the population (whose overall distribution provides the output).

-Also, the authors should explain the differences between the measurements used to obtain simulated and real results. The model was developed as a collection of single 'cells' with varied parameters. However, this work shows that aggregate fluorescence measurements of the sample suffice as opposed to a cell-by-cell evaluation (via flow cytometry, for example), which would otherwise elucidate more information about the population composition. Therefore, a real data overlay similar to Figure 2D (with an appropriate y axis) would be helpful to establish that the measurement choice yields a similar profile. This is an important point, given the nature of the analysis and nature of the data that can be brought to bear on the problem.

-Prioritization in the outcomes due to exposure events at the genetic, cell phenotype, and population level is needed. For example, 5 possible events occur, yet 4 DNA states are possible since Eb precludes further DNA alteration by Ea. Only 3 visual outputs result: off, RFP production, or GFP production. Nevertheless, while cells would remain off in a null event and Eba, Eba is still detectable at the population level. Output at each level is somewhat, but not wholly, representative of the state of response to the inducing event.

-Figures 3 and 4 (simulation and experiment) prove that measuring the prevalence of cell state Sab (green fluorescence) reveals the order and lapse between events, Eab or Eba. Can the authors better clarify to the reader how Eba can erroneously/stochastically result in predictable low population percentages of Sab?

-How do population kinetics affect the outcome of this system? Cells presumably must be metabolically active throughout the Δt interval in order to respond to both signals through integrase expression. Does the Δt limit correlate to growth rate changes? Is the system susceptible to bias if growth rates are altered due to the potential burden of protein expression?

-I am curious about the transition of a population from red fluorescence to green fluorescence. Is the depletion in red fluorescence primarily due to population growth, whereby the inverted DNA persists to eliminate RFP expression? If so, then to what extent does the initial subpopulation that experienced Ea exhibit both red (produced pre-inversion) and green fluorescence? What consequence would red subpopulations have on discrimination between Ea and Eab?

-Can the authors speculate on how the system would respond to inputs with

variable doses? There is no mention of this, yet this work provides the perfect vehicle to highlight this importance.

-Does the requirement that a total population response must be analyzed en-masse limit the system's "fieldability"?

-The final paragraph alludes to using this system as part of a synthetic community. Can the authors please develop this vision further for the reader? For example, to what extent would allocating multiple subpopulations with comparable logic gates further enrich the feedback output?

Minor Comments:

-The content of the work contrasts the notion of a single cell sensor, yet the text in the abstract does not provide the same contrast, especially by beginning with "Single cell bacterial sensors..." and thus misdirects the reader's initial mindset.

-Additionally, the wording of "transient chemical events" in the abstract seems inaccurate because the authors only explore one of the two events as a transient pulse while the other is maintained as a constant cue. To what extent would varying the duration of both chemical cues complicate the response behavior? This degree of predictability would be important for the system's "fieldability".

-Can a metric be provided in Figure 5C for the model/data correlation before and after parameter adjustment?

-Typos:

p. 3 "chemical chemical"

p. 11 "the different is"

p. 12 "transition into either of either of"

p. 12 "RPF and GFP"

p. 15 "source and concentration"

References: "Drew E."

1st Revision - authors' response

06 April 2016

Point-by-point Response to Reviewer 1

1. *Authors model the rate of integrase-mediated flipping as being linearly proportional to the amount of integrase, but this is an oversimplification. The authors themselves acknowledge that integrase-mediate recombination requires multimerization of the integrase (page 6, second to last paragraph). and so one would expect that recombination rate would be non-linear with respect to integrase concentration. Their simplification appears to be without justification.*

We agree with the reviewer that our treatment of flipping rate was oversimplified and have updated the model to include a term for the tetramerization of the integrase monomers (Eq. 2, pg. 5). We have also re-run all simulation results in the manuscript with the revised model. We searched the literature for data on serine integrases which would allow us to model flipping rates in a more sophisticated way. For our integrase A, TP901-1, we could not find any data. For our integrase B, Bxb1, we found data that it forms dimers first in solution, and that those dimers then bind the attB and attP sites (Ghosh et al, 2005). However, the literature stated that there was no evidence of cooperativity in binding of Bxb1 dimers to the attB and attP sites (Singh et al, 2014). That is to say, binding of a dimer at the attB site did not affect the probability of a dimer binding to attP. For yet a different serine integrase, PhiC31, which we did not use, there was evidence of cooperative binding of monomers to form dimers at its attB site, but not at its attP site (McEwan et al, 2009).

Given this diversity in integrase binding behavior, we chose to use the simplest possible model to capture the nonlinearity of integrase activity: we simply require that all four monomers must bind for integrase activity to be nonzero. This model, which gives the propensity of flipping in Eq. 2 (pg.5) does not qualitatively change the results of our simulations or our conclusions. This intuition for this is that in our system the production rates of the integrases are high and there is only a single DNA target so the four-monomer threshold is quickly reached (Appendix Figure S1). If the production rates were lower, or the number of DNA targets were higher, such that they could sequester a significant amount of bound integrase, then we would expect the addition of this tetramerization term to more significantly affect our conclusions.

Derivation of the tetramerization term can be found in Appendix Section 12.2.

2. *Authors use a degradation constant of .01/hr for the integrases in their model and say that it corresponds to a ~69 minute half life (page 6 main text, page 12 supplement), but this actually corresponds to a ~69 hour half life, which seems unrealistic.*

We have redefined the parameters to be on a more realistic time scale with degradation rate now $k_{\text{deg}} = 0.3\text{hr}^{-1}$ which equals a 2.3 hour half life (Manuscript pg. 6, 1st paragraph; Appendix Table S1, pg. 4). Since the fluorescent reporters are not degradation tagged, the

protein half-life largely depends on cell division, which is slow in minimal media (Appendix Fig. S7, pg.11).

The revised parameters are now:

Parameter	Value	Units
$k_{\text{prod}A,B}$	50	$(\mu\text{m}^3 \cdot \text{hr})^{-1}$
k_{deg}	0.3	hr^{-1}
$k_{\text{flip}A}$	0.4	hr^{-1}
$k_{\text{flip}B}$	0.4	hr^{-1}
$k_{\text{leak}A}$	0	$(\mu\text{m}^3 \cdot \text{hr})^{-1}$
$k_{\text{leak}B}$	0	$(\mu\text{m}^3 \cdot \text{hr})^{-1}$

Table 1: Revised parameters

3. *Authors use population-level fluorescence measurements in their experiments and therefore never directly assay the heterogeneity of their cell populations to see if they are dividing into different states as expected. To better validate that their circuit is performing as expected, I recommend that the authors use cytometry for high-throughput single-cell measurements instead of population-level measurements so that they can directly measure the fractional size of different fluorescent subpopulations. For example, the authors claim that integrating the DNA target sites for the integrases into the genome results in a single-copy per cell, but this may not be true when cells are actively replicating. How does this affect the behavior of their circuit at the single cell level?*

We agree with the reviewers that single cell data from flow cytometry would provide a more quantitative way to measure subpopulation fractions. We have fully re-visited the experiments for step inputs (Figure 4) and pulse inputs (Figure 7) and analyzed all the populations via flow cytometry. Analysis of flow cytometry data agrees with previous claims about population fractions, and all previous endpoint fluorescence data has been replaced with RFP and GFP population percentages.

Final population fractions for Figure 4C are now population percentages derived from exact cell counts ($\sim 100,000$ cells per population shown), and gated populations are now included in the new Expanded View Figure 1 (Figure EV1). This new data has been incorporated into the revised manuscript (Manuscript pg. 7, paragraphs 1–4). Additional flow cytometry data for Figure 4C can also be found in Appendix Fig. S8–10 (Appendix pg. 11-12).

Flow cytometry analysis of endpoint populations fully agreed with previous population fraction claims made with Biotek plate reader bulk fluorescence measurements, and provided additional interesting insights on non-fluorescent population fractions. Since time-course data (Figure 4A,B) was still measured via plate reader, we have included a comparison of plate reader fluorescence measurements to flow cytometry populations (Appendix Figure S12, pg. 14).

Flow cytometry was used to measure RFP, GFP, and non-fluorescent sub-populations for pulse experiments (Figure 7B,C), and all previous bulk fluorescence data has been replaced

with RFP and GFP population percentages (800,000 – 1 million cells per population were measured). Analysis of this new data is now incorporated into the manuscript (pg. 9, last two paragraphs). Visualization of subpopulations and fluorescence gating can now be seen in Extended View Figure 2 (EV2). (See also Appendix Figures S18–23, pg.20-25).

When cells are actively replicating, there may be more than one copy of the integration site per cell. However, single colony analysis of the resulting populations shows that individual colonies were monoclonal and maintained state even after repeated re-streaking (Appendix Figures S11 (pg.13), S24 (pg.26–27)).

4. *The authors should give more information about the growth of the cells during their experiments. Presumably the cells switch from exponential to stationary phase at some point during their experimental time courses. For example, if the second inducer is added closer or further away from stationary phase will it affect the way that the population divides into different states?*

Cells were grown in minimal media M9CA to ensure that the cells were in exponential phase during the entire time of the experiment (See Appendix Fig. S7, pg.11). The reviewer is correct in that the circuit behaves differently when the population reaches stationary phase. In general we have observed that the circuit is less responsive to inducers in late-log/stationary phase, and thus all of our induction times are done while the population is in the OD600 0.1 - 0.7 range. Appendix Fig. S7 shows growth curves from 0 – 30 hours. The final induction time occurs at $t = 10$ hours (OD 0.3), however the population does not reach stationary phase until $t = 25$ hours.

5. *Overall, more information in the experimental section would be useful. For example, how many replicates did the authors use in their experiments?*

Each experiment contains three independent replicates which are averaged. This information has been added to the Materials and Methods section, as well as many more additional details about starting OD, inducer concentrations, and growth conditions (Manuscript, pg. 12).

6. *With regard to the model fitting (figure 5C), authors should perform a more thorough analysis of the estimated parameters from the fit. For example, are these parameters a unique solution or is it possible that a completely different set of parameters could have given just as good of a fit? If the latter is the case, then the estimated set of parameters could be meaningless.*

In the revised manuscript, we have performed more thorough analysis of the estimated parameters (Appendix Figure S13, pg. 15), and revised Figure 5C to reflect this.

In Figure 5AB, we vary the parameters to gain a qualitative understanding of overall trends in behavior of the system. In particular, we learned that the population split at $\Delta t = 0$ h shifts monotonically downwards as the flipping efficiency of *intA* decreases relative to *intB*. Additionally, we observed that leaky expression of *intB* would take away from the maximum S_{ab} population fraction at $\Delta t = 8$ h, but also observed that measured leaky integrase expression was actually very low ($\leq 3\%$).

Based on flow cytometry data, we fixed $k_{\text{leak}A}$ at 1% and $k_{\text{leak}B}$ at 2% and focused on the tuning of $k_{\text{flip}A}$ and $k_{\text{flip}B}$. We ran E_{ab} simulations for a matrix of $k_{\text{flip}A}$ and $k_{\text{flip}B}$ values from 0.1 to 0.6 hr^{-1} (6 x 6 matrix) and found the mean squared error (MSE) for each simulation result as compared to experimental data (Appendix Figure S13B, pg. 15). The pair of values that generated the minimum MSE was $k_{\text{flip}A} = 0.2$ and $k_{\text{flip}B} = 0.3$.

7. *Authors claim that the Δt_{90} detection limit can be tuned by modulating the overall production rate of the integrases. The authors should show this experimentally. It seems like production rates could be tuned by mutating the ribosome binding sites of the integrases or the promoters from which they are expressed.*

We have clarified in the main text that the effect of modulating $k_{\text{prod}A,B}$ on Δt_{90} is a preliminary insight from simulations, and needs to be proven experimentally in future work (Manuscript, pg. 8, paragraph 4). We have also moved discussion of k_{prod^*} effect on Δt_{90} to the Appendix (pg. 16), and improved our analysis of simulation data (Appendix Figure S14, pg. 16).

Additionally, we have done some preliminary experiments to modulate production rates by decreasing the inducer concentrations of **a** (arabinose) and **b** (aTc) by half (0.005%/vol, 100ng/ml, respectively). These new data are included in Appendix Figure S12B (pg.16) and S13 (pg.17). However, the estimated Δt_{90} remains the same, and is consistent with simulation data for the saturation regime of integrase production. Creating a library new strains with varying RBSs is not within the scope of this study, as those systems would have very different parameters. Here, we wanted to focus more on what information could be obtained from population fractions.

8. *On page 12 paragraph 2 the authors say "Our numerical simulations predicted a complete overlap of the S_a curves (figure 6E), and the experimental results are consistent with those predictions, though there is some downwards drift with increasing Δt . This sentence contradicts itself. It suggests that fraction of cells in S_a is invariant to Δt even when the data (figure 7B) clearly shows a trend with Δt . The authors need to be more upfront about this incongruence between their model and the experimental data and suggest reasons why it might be happening and how it might be resolved. For example, is it an integrase leakiness issue?*

We have revised the text with improved discussion on these discrepancies between our model and experimental populations (Manuscript, pg. 10, paragraph 3). Our flow cytometry analysis shows < 3% maximum leakiness for either integrase so we don't believe this is a leaky expression issue. After careful analysis of the flow cytometry data, we believe the higher RFP in cases with lower Δt is due to unequal transitions rates between $S_o \rightarrow S_b$ and $S_a \rightarrow S_{ab}$. Though both transitions are mediated by intB, the differences in DNA configurations could explain the differences between S_a cell counts based on Δt . We have added detailed discussion of this in the main text (pg. 10, paragraph 3), with extensive new explanations and simulations in Appendix Section 9 (pg.28) and Appendix Figure S25 (pg.29). We also note that although there is this drift, different combinations of $(PW_b, \Delta t)$ still map to unique (S_a, S_{ab}) population fractions *in silico* and *in vivo*.

9. *In the experimental methods section, the authors forgot to put the source and concentration of chloramphenicol and kanamycin- instead it just reads "source and concentration".*

This has been corrected (Manuscript, pg. 12, last paragraph).

10. *In the discussion, the authors talk about how this system could be adapted to microRNA detection. It is unclear how this would work in the context of the current system - would the microRNAs be taken up into the bacteria, and if so, how? Or are the authors alluding to some in vitro system?*

In the current context, we would need to implement the temporal logic gate in mammalian cells and also re-design the system such that the mRNAs for the integrases are regulated via RNA hairpin structures. These toehold switches would form secondary structure hairpins that prevent translation unless activated by complementary RNAs (Green, A. et al, 2014). These switches can be rationally designed to respond to endogenous RNAs, and so we envision that we could design a proof-of-concept system that re-creates microRNAs. Detection of miRNAs would require implementation of the temporal logic gate in mammalian cells. Though recombinase-based synthetic circuits have not been shown in mammalian cells, serine integrases have been used quite effectively in a wide variety of mammalian cell types, primarily for genome editing and integration (Keravala *et al*, 2006; Xu *et al*, 2013). We have added additional text in the discussion section to clarify this (Manuscript pg. 11, paragraph 4).

11. *First sentence of paragraph 2 on page 3, the word "chemical" is written twice in a row. In the final paragraph of the introduction "E. coli" is not italicized.*

This has been corrected (Manuscript, pg. 3).

12. *On page 5 the statement that the attL and attR sites are "no longer recognized by the integrase" warrants a citation.*

A citation to (Ghosh *et al*, 2005) has been added (Manuscript, pg. 4, paragraph 5). In the cited study of Bxb1 recombination and directionality, the authors find that, "integrase alone does not promote excisive recombination using attL and attR as substrates."

13. *Authors claim that GFP and RFP provide "real-time" readout of the temporal logic gate, but they do not take into account the time delay it takes to express these proteins to appreciable levels for measurement.*

Yes, there is definitely a delay between when the DNA state switches and when the fluorescent proteins mature and built up to measurable concentration. In particular, our system has about a 2–4 hour delay before any fluorescence can be detected via the plate reader. However, even with the delay, we can observe different Δt inductions result in different GFP production slopes, which our model suggests may be because of competing reactions from S_o to S_a versus S_b states.

Furthermore, although we show time-course measurements as a way of understanding the system, we have made sure to only use our model to predict final endpoint distributions. We make the assumption that if the time of endpoint measurement is sufficiently long enough after the final induction time (at least 20 hours), that all cells in either of the RFP or GFP expressing states will be detectable via fluorescence.

14. *Why does GFP fold change peak over the experimental time course in Figure 4A?*

In the original Figure 4A, the GFP fold change had a peak during the experimental time course because there was leaky GFP expression in the *no inducer* control sample that increased over time at a rate that was different than the GFP production from S_{ab} state cells. We have re-done these experiments to collect flow cytometry data, so Fig. 4A has been replaced with new timecourse fluorescence data to show that all final populations were collected at steady state. To reduce confusion, the new data was normalized only by OD, not by the *no inducer* control. There is still a small residual peak in the revised Fig. 4A due to saturation of growth curves (Appendix Fig. S7, pg. 11).

15. *Is fluorescence being normalized by OD? What is the relative growth rate of cells exposed to different combinations of the inputs, and does this affect the measurements? It would be helpful to show the OD of the different curves and explicitly state how data is being processed in the methods / captions.*

Yes, fluorescence is being normalized by OD. The relative growth rate of the cells is constant regardless of inputs. We have added Appendix Figure S7 (pg.11) to show the relative growth curves.

16. *Do the E_{ba} measurements (Fig 4B and 4C) not have error bars, or are they just really small?*

The original E_{ba} measurements just had really small errorbars, however this figure has been replaced with flow cytometry data in the revised manuscript.

17. *Color choices for figure 4A and 4B make it hard to distinguish lines belonging to different Δt values.*

We agree the individual traces are difficult to distinguish. However, we felt that the color gradients were the best way to show the overall progression of endpoint populations with increasing Δt . We have added Appendix Figure S6 (pg. 10), which has the same curves as Figure 4 but with a more distinct color scheme so that it is easier to distinguish between individual curves.

18. *In Figure 6E, all of the lines are hidden behind the red-line because of overlap, is there a way to avoid this?*

Yes, all the lines overlap and are hidden behind the red-line. We have tried to resolve this by including the legend with all the lines, but also to increase the transparency of all the lines to hopefully show that all lines are overlapping.

19. *In Figure S2, I do not understand the plots showing the trajectories of cell state (So, Sa, Sb, Sab). Why are these plots labeled "copies per cell" on the y-axis and why do trajectories sometimes have values that are not Boolean over time? If the lines represent whether or not a cell is in a state, shouldn't they just be jumping between two Boolean values ("in state" versus "not in state")?*

The original stair plots for the state transition graphs may not have appeared boolean when the figure size was adjusted in the manuscript, but now have been corrected in Appendix Figure S2 (pg.6). The y-axis labels have also been relabeled for clarity. The integrase traces are labeled "Copies per cell" because although cell states are discrete, the numbers of integrase molecules are not and have a steady state molecular counts of $k_{\text{prod}*}/k_{\text{deg}}$ molecules.

Point-by-point Response to Reviewer 2

1. *More references on the topic of utilizing cell populations for classification of chemical/molecular input would be helpful. For example there is work on cell networks for conveying molecular communication that is highly relevant to this work.*

We have added additional references in the introduction that refer to theoretical and experimental work on population level understanding of cell responses. Specifically, we added references to work by Uhlendorf and colleagues that implemented model predictive control of populations (Uhlendorf J, et al, 2012) and systems identification (Ruess J, et al, 2015) by implementing stochastic cell models. Mathis and Ackermann have recently observed population-level memory of stress responses in *C. crescentus* (Mathis and Ackermann, 2016). These references have been added to the introduction (Manuscript pg. 3, paragraph 4).

2. *The hierarchical organization of the system, and its contributions to the output at each level, is a compelling feature. Thus, it would be helpful to strongly differentiate between the respective roles of the single cell (logic design, model of cell response parameters, etc) and the population (whose overall distribution provides the output).*

We strongly agree and have revised the manuscript to emphasize these differences. Specifically, Figure 3D in the manuscript now has a single-cell level versus population level chart (Manuscript, pg. 6, 2nd-to-last paragraph), and the additional information gained from population-level analysis is now a running theme in the paper. We believe this really focuses the paper and are very appreciative to the reviewer for bringing up this point.

3. *Also, the authors should explain the differences between the measurements used to obtain simulated and real results. The model was developed as a collection of single 'cells' with varied parameters. However, this work shows that aggregate fluorescence measurements of the sample suffice as opposed to a cell-by-cell evaluation (via flow cytometry, for example), which would otherwise elucidate more information about the population composition. Therefore, a real data overlay similar to Figure 2D (with an appropriate y axis) would be helpful to establish that the measurement choice yields a similar profile. This is an important point, given the nature of the analysis and nature of the data that can be brought to bear on the problem.*

We have addressed this by both re-running all experiments in the paper to do flow cytometry analysis (which gives us single cell counts) and also using that flow cytometry analysis to do a comparison with endpoint bulk fluorescence measurements. The addition of flow cytometry analysis has allowed us to replace the endpoint population measurements with actual population fractions (Figure 4C, Figure 7B,7C) and also to provide a basis of comparison with bulk fluorescence measurements (Appendix Figure S12, pg.14). We conclude based on this comparison that GFP bulk fluorescence measurements (which we still use for time-course traces in Figure 4A,4B) are representative of S_{ab} populations, while RFP bulk fluorescence measurements may not be as accurate since cells that spend more time in S_a also require more time after the state transition to fully dilute out all of the standing RFP molecules.

4. *Prioritization in the outcomes due to exposure events at the genetic, cell phenotype, and population level is needed. For example, 5 possible events occur, yet 4 DNA states are possible since E_b precludes further DNA alteration by E_a . Only 3 visual outputs result: off, RFP production, or GFP production. Nevertheless, while cells would remain off in a null event and E_{ba} , E_{ba} is still detectable at the population level. Output at each level is somewhat, but not wholly, representative of the state of response to the inducing event.*

We completely agree, and have added a chart in Figure 3D of the manuscript to clarify all of the possible single-cell genetic states, single-cell fluorescent outputs, and population-level distributions with each possible event (Manuscript pg.6, 2nd-to-last paragraph).

5. *Figures 3 and 4 (simulation and experiment) prove that measuring the prevalence of cell state S_{ab} (green fluorescence) reveals the order and lapse between events, E_{ab} or E_{ba} . Can the authors better clarify to the reader how E_{ba} can erroneously/stochastically result in predictable low population percentages of S_{ab} ?*

Using flow cytometry results, we can now identify about 0.5-3% leaky expression of both integrase A and B, which can result in non-zero population percentages of S_{ab} even with high Δt . Specifically, we can examine the *a only* and *b only* population quadrants in Figure EV1 and EV2 to find overall populations that result from leaky behavior. This is discussed in the main text (pg. 7, paragraphs 3–4), as well as in the figure legends for Figure EV1 and EV2.

6. *How do population kinetics affect the outcome of this system? Cells presumably must be metabolically active throughout the delta t interval in order to respond to both signals through integrase expression. Does the delta t limit correlate to growth rate changes? Is the system susceptible to bias if growth rates are altered due to the potential burden of protein expression?*

Populations kinetics are indeed important to the outcome of this system. Unfortunately, the system is subject to the same limitations as other synthetic circuits that rely on the Ptet and PBAD inducible promoters, which are optimally active in exponential phase. We have taken precautions to ensure that the Δt limit does *not* correlate to growth rate changes due to later

induction times. In particular, we use minimal M9CA media with glycerol to extend the time before stationary phase (and to allow integrase molecules to build up in each cell) and ensure that all experimental conditions have the same growth curves (Appendix Figure S7, pg. 10).

Yes, the system is definitely susceptible to bias with altered growth rates due to protein expression load. Flow cytometry analysis of the final distributions showed that final distributions of S_a or S_{ab} cells had a ceiling of 60-70% of the population (Figure 4C, Figure 7B,C). We had two possible hypotheses: (1) *intB* is extremely leaky and automatically converting 30% of the population into S_b cells, or (2) the non-fluorescent states S_o and S_b have a slight growth advantage and so are overrepresented in the final populations due to faster proliferation. When we did single colony analysis of the non-fluorescent cells for each of the populations (Appendix Figure S11, pg. 13), we find that populations growth for 40 hours with no inducers were almost 100% still in state S_o , which ruled out hypothesis #1. Therefore, we concluded that the non-fluorescent states are overrepresented due to growth advantages, even with single-gene integrants of the fluorescent proteins (though with very strong constitutive promoter p7 + very strong bicistronic ribosomal binding sites, BCD1 and BCD2). This is something that is not accounted for in our model, and we intend to give this further exploration in future studies, particularly with multi-strain consortia.

7. *I am curious about the transition of a population from red fluorescence to green fluorescence. Is the depletion in red fluorescence primarily due to population growth, whereby the inverted DNA persists to eliminate RFP expression? If so, then to what extent does the initial subpopulation that experienced E_a exhibit both red (produced pre-inversion) and green fluorescence? What consequence would red subpopulations have on discrimination between E_a and E_{ab} ?*

Yes, since the proteins are not degradation tagged, the depletion of red fluorescence depends on population growth. This means that the longer a cell spends on state S_a , the longer it takes to dilute out all of the RFP molecules (Appendix Figure S4A, pg. 8). With bulk fluorescence measurements, it can be difficult to distinguish between S_a cells versus S_{ab} cells that have not yet diluted out all RFP yet. However, with flow cytometry this distinction is very clear when we plot RFP versus GFP fluorescence for each cell (Figure EV1A). Using flow cytometry analysis of single cells, we can divide the populations into four quadrants: Q1 (GFP only, S_{ab}), Q2 (GFP and RFP, S_{ab} with undiluted RFP), Q3 (RFP only, S_a), and Q4 (non-fluorescent, $S_o + S_b$).

In Figure EV1, we show these population quadrants for the step induction Δt experiments (the same ones shown in Figure 4). In these experiments, after Δt , both inducers remain in the media for up to 20 hours after induction. Thus, we would expect no S_a cells to remain in the final populations, and indeed we see < 3% of Q3 (RFP only, S_a) in all final populations except for the *a only* condition, which is 58% Q3 cells. In these experiments, we do see a high percentage of cells in the Q2 transition state, where the cells have genetically switched to S_{ab} but still contain measurable levels of RFP. This is not surprising since the cells are slow growing in minimal media and do slow down in growth as they approach OD 1 by the end of the experiment.

High percentages of cells in Q2 (GFP and RFP, S_{ab} with undiluted RFP) would be a concern

in the pulse width experiments (Figure 6,7) since RFP populations are a critical part of determining unique combinations of Δt and PW_b . When we did flow cytometry analysis of these populations, however, Q2 populations were $< 3\%$ for all experimental conditions (Figure EV2). This indicates RFP is still a reliable measure of S_a cells for these experiments. We believe this is because inducer **b** has a much shorter pulse width in these experiments (6 hours maximum) and so fewer cells are transitioning. In addition, the pulse is achieved through dilution in to new media + inducer **a** and additional growth, allowing for much more cell growth and RFP dilution than the step input experiment.

8. *Can the authors speculate on how the system would respond to inputs with variable doses? There is no mention of this, yet this work provides the perfect vehicle to highlight this importance.*

In this study, we wanted the inducers to be only "on" or "off" since we wished to focus our analysis on separation time and pulse width. Clearly, variable doses are important, however, we did not want to add additional variables since Ptet and PBAD have different non-linear induction curves, and so we settled on one concentration for arabinose and one concentration of aTc.

We did investigate the effect of whether cutting inducer concentrations in half would change the Δt_{90} limit by slowing protein production (Appendix Figure S14, pg. 16; S15, pg. 17), and observed a more graded effect on S_{ab} fractions as a function of Δt .

We believe the effect of variable doses would certainly affect overall population distributions and hope to do further study on amplitude versus frequency modulation of the system. For instance, how do population distributions compare when subjected so a slow ramp up versus multiple short pulses of high concentration? These are very interesting questions to pursue in the future, but are outside of the scope of this study.

9. *Does the requirement that a total population response must be analyzed en-masse limit the system's "fieldability"?*

We do not believe that the entire population needs to be analyzed to gain accurate insight of the total distribution. While we did not do a detailed study on the minimum cells needed to reflect the total population, our analysis of bulk (entire population) fluorescence, versus flow cytometry (100,000 - 1 million cells per population), versus single colonies (~ 50 colonies per population) all show fairly consistent RFP vs GFP vs non-fluorescent distributions.

Initially, we did all of our experimental work with bulk fluorescence measurements (500ul cultures, grown to about OD 1) but this did not provide single cell data or a reliable measure of the non-fluorescent population fractions. During revisions of the manuscript, upon reviewer recommendation, we extended our analysis to flow cytometry measurements. For flow cytometry, we analyzed 100,000 to 1 million cells per population (each individual Δt , PW_b condition was a separate population), which for 10^9 cells per ml is less than 0.1% of the total population (Fig. EV1, Fig. EV2). Flow cytometry results led us to become curious as to the genetic identity of the non-fluorescent population fraction, and during this process, we plated a very small fraction of the original experimental cultures, then picked about 50

random single colonies for PCR and sequencing analysis (Appendix Figure S11, pg. 13; Appendix Figure S24, pg.26–27)). When we counted the distributions of these 50 single colonies, they closely approximated the flow cytometry distributions we measured.

Thus, we do not believe that cell counts are going to be the limiting factor to this system's fieldability. Though more cells will provide higher resolution, as long as some small percentage of the original population can be recovered, we think this will be sufficient to determine population fractions.

10. *The final paragraph alludes to using this system as part of a synthetic community. Can the authors please develop this vision further for the reader? For example, to what extent would allocating multiple subpopulations with comparable logic gates further enrich the feedback output?*

Once we have engineered a synthetic "stem cell" with precise ways of creating sub-population distributions, we can begin to start understanding ways to add feedback and control to these distributions. Rather than designing a system with certain output, we could design systems with a certain distribution of outputs. This greatly expands our capability to design synthetic systems that have controllable distributions as outcomes, not just digital on/off phenotypes. Furthermore, we can then begin to develop frameworks for understanding the role of feedback and control theory in modulating these sub-populations given different starting distributions. We have added this discussion to the last paragraph of the Discussion section (Manuscript, pg. 12, first paragraph).

Reviewer 2 Minor Comments:

1. *The content of the work contrasts the notion of a single cell sensor, yet the text in the abstract does not provide the same contrast, especially by beginning with "Single cell bacterial sensors...", and thus misdirects the reader's initial mindset.*

We have revised the abstract to emphasize this contrast (Manuscript, pg. 2).

2. *Additionally, the wording of "transient chemical events" in the abstract seems inaccurate because the authors only explore one of the two events as a transient pulse while the other is maintained as a constant cue. To what extent would varying the duration of both chemical cues complicate the response behavior? This degree of predictability would be important for the system's "fieldability".*

Although one of the events is used as reference, we explore a variety of transient inducer **b** pulses, so we believe it is valid to say "transient chemical events." Yes, varying the duration of both chemical cues would complicate the response behavior, though we believe that given a specific application, we could use the Markov model to predict approximate distributions first.

3. *Can a metric be provided in Figure 5C for the model/data correlation before and after parameter adjustment?*

We have added mean squared error (MSE) as a metric for comparison of original simulation results to the simulation results with adjusted parameters (Appendix Figure S13, pg.15). Figure 5C has been revised to show the simulation with initial parameters, experimental data, and model with revised parameters (Manuscript, pg. 8, paragraphs 2–3). We ran E_{ab} simulations for a matrix of $k_{\text{flip}A}$ and $k_{\text{flip}B}$ values from 0.1 to 0.6 hr^{-1} (6 x 6 matrix) and found the mean squared error (MSE) for each simulation result as compared to experimental data (Appendix Figure S13B, pg.15). The pair of values that generated the minimum MSE was $k_{\text{flip}A} = 0.2$ and $k_{\text{flip}B} = 0.3$.

4. *Typos*: We have fixed all of these.

p. 3 "chemical chemical"

p. 11 "the different is"

p. 12 "transition into either of either of"

p. 12 "RPF and GFP"

p. 15 "source and concentration"

References: "Drew E."

Corresponding Author Name: Victoria Hsiao

Manuscript Number: MSB-15-6663